# Scaling Transformers for End-to-End Discrete Audio Tokenization

**Yitian Gong**[1 2 3]  **Kuangwei Chen**[1 2 3]  **Zhaoye Fei**[1 3]  **Xiaogui Yang**[3]  **Ke Chen**[1 2 3]  **Yang Wang**[1 3]
**Kexin Huang**[1 3]  **Mingshu Chen**[1 2 3]  **Ruixiao Li**[1 2 3]  **Qinyuan Cheng**[1 3]  **Shimin Li**[3]  **Xipeng Qiu**[1 2 3]

## Abstract

Discrete audio tokenizers are fundamental to empowering large language models with native audio processing and generation capabilities. Despite recent progress, existing approaches often rely on pretrained encoders, semantic distillation, or heterogeneous CNN-based architectures. These designs introduce fixed inductive biases that limit reconstruction fidelity and hinder effective scaling. In this paper, we argue that discrete audio tokenization should be learned fully end-to-end using a homogeneous and scalable architecture. Based on this perspective, we propose **TAC**, a Transformer-based audio tokenizer that jointly optimizes the encoder, quantizer, and decoder from scratch for high-fidelity reconstruction of general audio. We show that a simple, fully end-to-end learned tokenizer built from homogeneous, causal Transformer blocks scales gracefully and supports high-fidelity reconstruction across diverse audio domains. Across speech, sound, and music, the proposed tokenizer consistently outperforms prior codecs over a wide range of bitrates, while exhibiting predictable improvements with increased scale. Notably, leveraging TAC's discrete tokens, we develop the first purely autoregressive TTS model that surpasses prior non-autoregressive and cascaded systems. Furthermore, TAC enables competitive ASR performance without auxiliary encoders. Our findings position TAC as a unified, scalable interface for the next generation of native audio foundation models.

## 1. Introduction

Recent advances in large language models ([Brown et al.,](#) [2020;](#) [Touvron et al., 2023;](#) [Achiam et al., 2023;](#) [Hurst et al.,](#)

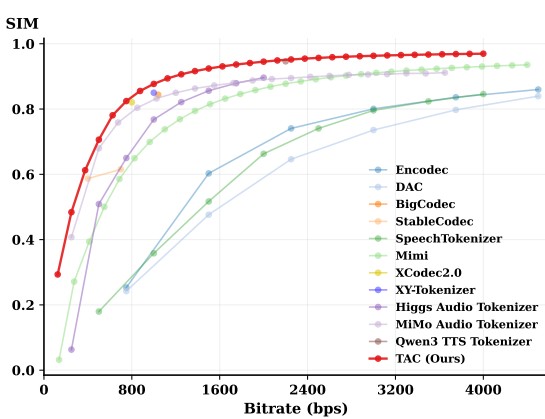

*Figure 1.* Audio reconstruction quality comparison between TAC and open-source audio tokenizers. At comparable bitrates (0–4000 bps), TAC consistently achieves higher SIM scores than all other compared tokenizers.

[2024;](#) [Dubey et al., 2024;](#) [Yang et al., 2024a;](#) [2025;](#) [Guo et al., 2025](#)) have demonstrated the effectiveness of autoregressive modeling over discrete token sequences. By providing a unified discrete interface, text tokenizers ([Sennrich et al., 2016;](#) [Kudo & Richardson, 2018](#)) allow large language models to operate directly on raw text, serving as the foundation upon which compression, understanding, generation, and in-context learning capabilities emerge within a single autoregressive modeling framework. Extending this paradigm to audio requires a unified discrete audio tokenizer that can serve as a native interface for autoregressive modeling ([Borsos et al., 2023;](#) [Agostinelli et al., 2023;](#) [Yang et al., 2024b;](#) [Zhang et al., 2025b](#)).

Unlike text, audio contains both fine-grained acoustic details and long-range structure, making discrete tokenization more challenging ([Borsos et al., 2023](#)). A unified audio tokenizer should enable high-fidelity reconstruction of diverse audio signals while remaining compatible with autoregressive sequence modeling ([Défossez et al., 2024;](#) [Li et al., 2025c;](#) [Zhang et al., 2025b](#)). Existing approaches typically address these requirements through pretrained encoders ([Hsu et al., 2021;](#) [Chung et al., 2021;](#) [Radford et al., 2023;](#) [Ye et al., 2025a;](#) [Gong et al., 2025;](#) [Li et al., 2025a](#)), multistage training pipelines ([Wu et al., 2023;](#) [Welker et al., 2025;](#) [Zhang et al., 2025b](#)), or architecture-specific inductive biases ([Zeghidour et al., 2021;](#) [Défossez et al., 2022;](#) [Kumar et al., 2023](#)), achieving strong performance under particular

---

[1]Fudan University [2]Shanghai Innovation Institute [3]MOSI Intelligence. Correspondence to: Xipeng Qiu <xpqiu@fudan.edu.cn>.

*Proceedings of the $43^{rd}$ International Conference on Machine Learning*, Seoul, South Korea. PMLR 306, 2026. Copyright 2026 by the author(s).

*Table 1.* Comparison of representative audio tokenizers with respect to architectural design and functional capabilities. ✓ indicates support, ✗ indicates not supported, and '–' indicates not specified. Trans. denotes Transformer, and Hybrid denotes a hybrid architecture combining CNN and Transformer. End-to-end optimize indicates whether all modules are jointly optimized under a unified objective.

| Model | Frame rate | Encoder Arch. | Decoder Arch. | Stream–ing | Variable Bitrate | Semantic rich | Reconstruction | | | Pretrained encoder free | End-to-end optimize |
|---|---|---|---|---|---|---|---|---|---|---|---|
| | | | | | | | Speech | Sound | Music | | |
| Encodec | 75 | CNN | CNN | ✓ | ✓ | ✗ | ✓ | ✓ | ✓ | ✓ | ✓ |
| DAC | 75 | CNN | CNN | ✗ | ✓ | ✗ | ✓ | ✓ | ✓ | ✓ | ✓ |
| SpeechTokenizer | 50 | CNN | CNN | ✗ | ✓ | ✓ | ✓ | ✗ | ✗ | ✗ | ✗ |
| Mimi | 12.5 | Hybrid | Hybrid | ✓ | ✓ | ✓ | ✓ | ✓ | ✓ | ✗ | ✗ |
| BigCodec | 80 | CNN | CNN | ✗ | ✗ | ✗ | ✓ | ✗ | ✗ | ✓ | ✓ |
| Stable Codec | 25 | Hybrid | Hybrid | ✗ | ✓ | ✗ | ✓ | ✗ | ✗ | ✓ | ✓ |
| XCodec2.0 | 50 | Hybrid | Hybrid | ✗ | ✗ | ✓ | ✓ | ✗ | ✗ | ✗ | ✗ |
| XY-Tokenizer | 12.5 | Hybrid | Hybrid | ✗ | ✗ | ✓ | ✓ | ✗ | ✗ | ✗ | ✗ |
| DualCodec | 12.5 | Hybrid | CNN | ✗ | ✓ | ✓ | ✓ | ✗ | ✗ | ✗ | ✗ |
| Higgs Audio Tokenizer | 25 | Hybrid | CNN | ✗ | – | ✓ | ✓ | ✓ | ✓ | ✗ | ✗ |
| MiMo Audio Tokenizer | 25 | Hybrid | Hybrid | ✗ | ✓ | ✓ | ✓ | ✓ | ✓ | ✓ | ✗ |
| Qwen3 TTS Tokenizer | 12.5 | Hybrid | Hybrid | ✓ | – | ✓ | ✓ | – | – | ✗ | ✗ |
| TAC (Ours) | 12.5 | Trans. | Trans. | ✓ | ✓ | ✓ | ✓ | ✓ | ✓ | ✓ | ✓ |

design choices. However, such designs introduce additional dependencies and architectural constraints that make it difficult to scale models, data, and quantization capacity in a unified manner. From this perspective, we draw inspiration from the success of large language models, where simple and efficient architectures trained on large-scale data have proven critical for achieving strong performance (Kaplan et al., 2020; Henighan et al., 2020). We posit that enabling an audio tokenizer to reach a higher performance ceiling similarly requires a simple and scalable architecture that can be trained end-to-end on large amounts of data. Such a design emphasizes joint optimization and scale, while minimizing reliance on external priors, pretrained components, or complex architectural heuristics.

In this work, we propose a fully end-to-end audio tokenizer that serves as a unified discrete interface for autoregressive audio language models. Our tokenizer, referred to as a Transformer-based Audio Codec (TAC), operates at a 24 kHz sampling rate with a low token frame rate of 12.5 Hz, and jointly optimizes the encoder, quantizer, decoder, and discriminator within a single training pipeline, without relying on pretrained encoders, distillation, or separate optimization of individual components. Both the encoder and decoder are built entirely from causal Transformer blocks, resulting in a simple and scalable architecture that is naturally aligned with autoregressive modeling (Vaswani et al., 2017). All components of TAC are designed to operate in a streaming manner, enabling low-latency inference and consistent training–inference behavior (Zeghidour et al., 2021; Défossez et al., 2022; 2024).

By scaling large amounts of paired audio–text data, TAC learns discrete representations that are both structurally rich and acoustically expressive, while remaining robust across a wide range of bitrates. As a result, the tokenizer achieves high-quality reconstruction of general audio, including speech, sound, and music, from very low to high bitrate regimes, providing a strong lower bound and a high upper bound for downstream audio language models.

Across speech, sound, and music, TAC achieves state-of-the-art reconstruction quality at all evaluated bitrates. Leveraging its discrete tokens, we further introduce a purely autoregressive text-to-speech model with a Progressive Sequence Dropout training strategy, which naturally exploits the tokenizer's robustness across bitrates and, for the first time, enables a fully autoregressive discrete TTS system (Liao et al., 2024; Ye et al., 2025b; Wang et al., 2025; Xie et al., 2025) to outperform prior non-autoregressive (Eskimez et al., 2024; Chen et al., 2025b) and cascaded approaches (Betker, 2023; Wang et al., 2023; Anastassiou et al., 2024; Du et al., 2024; Wang et al., 2024; Zhang et al., 2025a; Zhou et al., 2025a; Cui et al., 2025). In addition, TAC supports competitive automatic speech recognition performance without requiring an auxiliary audio encoder, matching or exceeding models that rely on dedicated audio encoders combined with large language models (Chu et al., 2024; Liu et al., 2025; Xu et al., 2025c;a;b). Together, these results demonstrate that TAC provides a scalable and effective foundation for audio compression, understanding, and generation within a unified autoregressive framework.

Our contributions can be summarized as follows:

- We propose TAC, a fully end-to-end Transformer-based audio tokenizer that serves as a unified discrete interface for autoregressive audio language models. TAC supports low-frame-rate and variable-bitrate tokeniza-

tion, enables high-fidelity reconstruction of general audio including speech, sound, and music, and demonstrates the effectiveness of jointly optimizing and scaling the encoder, quantizer, and decoder for discrete audio tokenization.

- Leveraging TAC, we introduce the first purely autoregressive discrete text-to-speech system that achieves state-of-the-art performance, surpassing prior non-autoregressive and cascaded TTS approaches. We further propose a **Progressive Sequence Dropout** training strategy that enables variable-bitrate generation within the same autoregressive model.

- We demonstrate that TAC provides a unified and effective foundation for audio compression, understanding, and generation, achieving state-of-the-art reconstruction quality across bitrates and strong downstream performance.

**Conflict of Interest Disclosure.**  Some authors are affiliated with MOSI Intelligence, an organization that conducts research and development in audio and language modeling. The authors disclose this affiliation as a potential financial conflict of interest. No other financial conflicts of interest are declared.

## 2. Rethinking Discrete Audio Tokenization for Autoregressive Audio Modeling

We rethink discrete audio tokenization from the perspective of autoregressive audio language modeling. Analogous to text tokenizers in large language models, a discrete audio tokenizer should serve as a native interface that bridges raw audio signals and autoregressive sequence modeling (Zhang et al., 2023b; Défossez et al., 2024; Zhang et al., 2025b). This viewpoint places stringent requirements on the structure, representation capacity, and scalability of the tokenizer, beyond traditional objectives of audio compression or reconstruction.

From this perspective, we identify several key design principles that an audio tokenizer must satisfy in order to effectively support autoregressive audio language models.

**Unified Audio Representation.**  A tokenizer should provide a unified discrete representation capable of modeling and reconstructing diverse audio domains, including speech, sound, and music. Crucially, the resulting tokens should preserve both fine-grained acoustic information and semantic structure, enabling them to function as a meaningful sequence for autoregressive modeling rather than merely a compressed signal.

**Simplicity and Scalability.**  To enable efficient scaling with model capacity, data, and computation, the tokenizer architecture should remain simple and homogeneous. Excessive architectural heterogeneity or reliance on specialized components can introduce fixed bottlenecks that hinder joint scaling and limit the effectiveness of large-scale training.

**Causality.**  For compatibility with autoregressive generation and low-latency inference, tokenization should be strictly causal, ensuring that each token is computed without access to future audio context. This property aligns the tokenizer with the operational constraints of autoregressive audio language models and avoids discrepancies between training and inference.

**Low Frame Rate and Bitrate Robustness.**  An effective audio tokenizer should operate at a low frame rate to reduce downstream sequence modeling complexity, while remaining robust across a wide range of bitrates. Such flexibility allows a single tokenizer to support diverse downstream tasks, including audio reconstruction, understanding, and generation, without requiring task-specific redesign.

## 3. Transformer-based Audio Tokenizer (TAC)

### 3.1. Homogeneous Architecture for Scalable Audio Tokenization

A central design goal of TAC is to enable scalable audio tokenization that can seamlessly integrate with large autoregressive and multimodal foundation models. To this end, we adopt a *CNN-free* architecture that is built entirely upon causal Transformer blocks, as illustrated in Figure 2. Compared to prior neural audio codecs that rely heavily on convolutional inductive biases or hybrid CNN–Transformer designs, our approach deliberately minimizes architectural specialization, favoring simplicity, uniformity, and scalability.

**Fully Transformer-based encoder–decoder.**  Both the encoder and decoder in TAC are implemented as stacks of causal Transformer blocks, forming a CNN-free architecture and enabling streaming encoding and decoding. TAC operates directly on raw audio waveforms at both the input and output, avoiding intermediate signal representations such as mel-spectrograms. The input waveform is first *patchified* into a sequence of fixed-dimensional vectors and processed by the causal Transformer encoder. To progressively compress long audio sequences into a compact representation, we insert patchify operations between Transformer blocks, which gradually reduce the temporal resolution. As a result, the encoder maps 24 kHz waveforms into discrete token sequences at an average rate of 12.5 frames per second. The decoder mirrors this process in reverse, reconstructing the

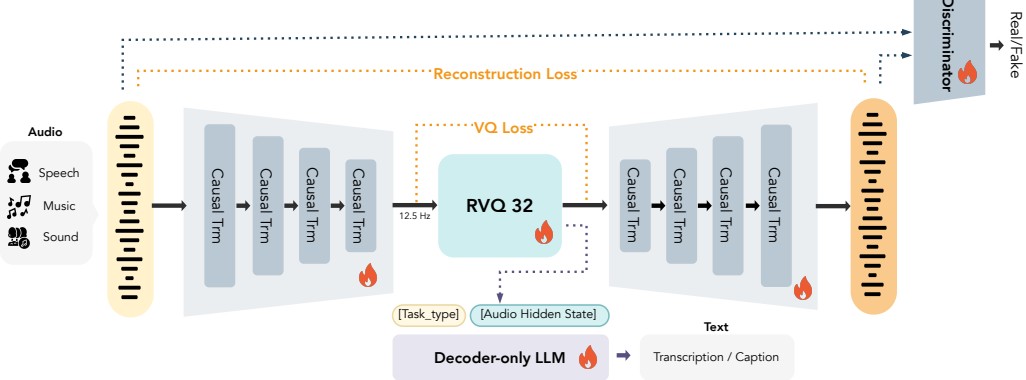

*Figure 2.* Architecture of TAC. Both the encoder and decoder are built upon causal Transformers. All components, including the encoder, quantizer, decoder, causal language model, and discriminator, are optimized jointly in an end-to-end manner.

waveform from discrete tokens in a fully causal manner. Further implementation details are provided in Appendix A.

**Scalable residual vector quantization.** For discretization, we employ residual vector quantization (RVQ). To support robust modeling across a wide range of bitrates, we adopt $N_q = 32$ residual quantization layers and enable quantizer dropout during training. This variable-bitrate design directly facilitates the controllable audio generation framework introduced later.

### 3.2. Unified Audio Modeling

We use multi-task learning to enable TAC to achieve both strong alignment with text and high-quality audio reconstruction.

**Semantic Modeling via Audio-to-Text Tasks.** To encourage the token representation to be semantically rich and aligned with text-based language modeling, we incorporate an auxiliary audio-to-text objective. Specifically, we employ a 0.5B-parameter decoder-only LLM (Yang et al., 2024a) and condition it on the representations produced by TAC. Concretely, we feed the hidden states from the quantizer output into the LLM, which then autoregressively predicts textual tokens. We consider a diverse set of audio-to-text tasks, including automatic speech recognition (ASR), multi-speaker ASR, and audio captioning. For audio samples that are paired with textual annotations, we apply the corresponding semantic modeling objective. Each task is specified by a fixed task tag $\mathcal{T}$, which is prepended to the LLM input. The semantic objective is optimized using a standard cross-entropy loss:

$$\mathcal{L}_{\text{sem}} = -\sum_{t=1}^{|\mathbf{s}|} \log p_{\theta_{\text{LLM}}}\left(\mathrm{s}_t \mid \mathcal{T}, \mathbf{q}, \mathrm{s}_{<t}\right), \quad (1)$$

where $\mathbf{s} = (\mathrm{s}_1, \ldots, \mathrm{s}_{|\mathbf{s}|})$ denotes the target text token sequence, $\mathbf{q}$ denotes the sequence of quantized audio representations produced by TAC, $\mathcal{T}$ is a task-specific prompt token, and $\theta_{\text{LLM}}$ are the parameters of the causal language model.

**Quantizer Optimization.** For training simplicity and stability, each quantization layer in TAC adopts factorized vector quantization (Kumar et al., 2023), where codebooks are directly optimized via gradient descent, without relying on additional codebook update mechanisms. We incorporate a commitment loss and a codebook loss to jointly optimize the encoder and the codebook entries:

$$\mathcal{L}_{\text{cmt}} = \sum_{c=1}^{N_q} \|\mathbf{z}_c - \mathrm{sg}(q_c(\mathbf{z}_c))\|_2^2, \quad (2)$$

$$\mathcal{L}_{\text{code}} = \sum_{c=1}^{N_q} \|\mathrm{sg}(\mathbf{z}_c) - q_c(\mathbf{z}_c)\|_2^2, \quad (3)$$

where $\mathbf{z}_c$ denotes the input to the $c$-th quantization layer, $q_c(\mathbf{z}_c)$ is the corresponding quantized output, $N_q$ is the number of quantizers, and $\mathrm{sg}(\cdot)$ denotes the stop-gradient operator (Van Den Oord et al., 2017).

**Acoustic Modeling via Reconstruction Tasks.** To ensure high-fidelity and domain-robust audio reconstruction, we adopt a multi-scale mel-spectrogram loss:

$$\mathcal{L}_{\text{rec}} = \sum_{i=5}^{11} \|S_{2^i}(\mathbf{x}) - S_{2^i}(\hat{\mathbf{x}})\|_1, \quad (4)$$

where $S_{2^i}(\cdot)$ denotes the mel-spectrogram computed using a normalized short-time Fourier transform (STFT) with window size $2^i$ and hop size $2^{i-2}$. Here, $\mathbf{x}$ is the ground-truth waveform and $\hat{\mathbf{x}}$ is the reconstructed waveform generated by the decoder.

**Adversarial Training.** To further improve reconstruction fidelity and perceptual quality, we employ adversarial training with multiple discriminators. Specifically, we adopt the discriminator architecture and training objectives, including the adversarial loss, feature matching loss and discriminator loss, following XY-Tokenizer (Gong et al., 2025).

**Overall Training Objective.** The overall generator objective is a weighted combination of all loss terms:

$$\mathcal{L}_{\mathrm{G}} = \lambda_{\mathrm{sem}}\mathcal{L}_{\mathrm{sem}} + \lambda_{\mathrm{rec}}\mathcal{L}_{\mathrm{rec}} + \lambda_{\mathrm{cmt}}\mathcal{L}_{\mathrm{cmt}}$$
$$+ \lambda_{\mathrm{code}}\mathcal{L}_{\mathrm{code}} + \lambda_{\mathrm{adv}}\mathcal{L}_{\mathrm{adv}} + \lambda_{\mathrm{feat}}\mathcal{L}_{\mathrm{feat}}, \quad (5)$$

where $\mathcal{L}_{\mathrm{adv}}$ and $\mathcal{L}_{\mathrm{feat}}$ denote the adversarial and feature matching losses defined in XY-Tokenizer (Gong et al., 2025). $\lambda_{\mathrm{sem}}, \lambda_{\mathrm{rec}}, \lambda_{\mathrm{cmt}}, \lambda_{\mathrm{code}}, \lambda_{\mathrm{adv}}, \lambda_{\mathrm{feat}}$ are scalar hyperparameters controlling the relative contribution of each loss term.

All components of TAC, including the encoder, quantizer, decoder, and discriminators, are optimized jointly in an end-to-end manner. By scaling large amounts of audio data, TAC learns to achieve both high-fidelity reconstruction of general audio and semantically rich discrete representations, without relying on pretrained encoders or external semantic teachers.

### 3.3. Bitrate Controllable Audio Modeling

**End-to-end variable-bitrate autoregressive speech generation.** Building on TAC, we construct a fully end-to-end, purely autoregressive speech generation model that supports variable-bitrate synthesis. The model directly generates speech from text tokens and a speaker prompt by predicting TAC RVQ tokens at a controllable depth, without requiring semantic disentanglement (Zhang et al., 2023b; 2024; 2025c) or cascading multiple generative models (Du et al., 2024; Anastassiou et al., 2024; Cui et al., 2025). By leveraging TAC as a unified discrete interface, both linguistic content and acoustic information are modeled within a single autoregressive framework.

**Autoregressive modeling over RVQ tokens.** Since TAC represents audio using residual vector quantization (RVQ), we adopt the **Temporal Transformer + Depth Transformer** architecture (Défossez et al., 2024) for multi-stream autoregressive modeling. The Temporal Transformer captures long-range dependencies along the temporal dimension, while the Depth Transformer models the coarse-to-fine residual structure across RVQ layers within each time step. Under this design, each RVQ token conditions only on tokens from previous time steps and on preceding RVQ layers at the current time step, ensuring strict causality without information leakage.

**Progressive Sequence Dropout.** To enable robust generation across a wide range of bitrates within a single model,

we propose *Progressive Sequence Dropout*, a simple yet effective training strategy that requires **no architectural modifications or additional parameters**. During training, dropout is activated with probability $p$. When activated, we uniformly sample a prefix length $K \in \{1, \dots, N_q - 1\}$, where $N_q$ denotes the total number of RVQ layers, and discard RVQ tokens from layers $K+1$ to $N_q$. Otherwise, all RVQ layers are retained. This strategy exposes the model to truncated RVQ prefixes during training, where a prefix length is randomly sampled independently for each training sample, encouraging the model to learn conditional generation under varying bitrates.

**Prefix definition.** We introduce a Bernoulli random variable

$$z \sim \mathrm{Bernoulli}(p), \quad (6)$$

where $z = 1$ indicates that Progressive Sequence Dropout is applied and $z = 0$ otherwise. When $z = 1$, the prefix length $K$ is sampled uniformly as described above; when $z = 0$, we set $K = N_q$. The effective number of active RVQ layers is then defined as

$$\hat{K} = (1 - z) N_q + z K. \quad (7)$$

**Global input aggregation and training objective.** Let $\mathbf{q}_{t,k}$ denote the RVQ token at time step $t$ and layer $k$, and let $\mathrm{Emb}_k(\cdot)$ denote the embedding lookup table for the $k$-th RVQ codebook. For each time step $t$, the speech input to the Temporal Transformer is constructed by aggregating the embeddings of the first $\hat{K}$ RVQ layers:

$$\tilde{\mathbf{e}}_t = \sum_{k=1}^{\hat{K}} \mathrm{Emb}_k(\mathbf{q}_{t,k}). \quad (8)$$

The Temporal Transformer processes the resulting acoustic embedding sequence $\{\tilde{\mathbf{e}}_t\}_{t=1}^{T}$ using a causal attention mask along the temporal dimension.

The Depth Transformer predicts RVQ tokens autoregressively along the depth dimension. The training loss is computed only over the retained RVQ prefix:

$$\mathcal{L} = -\sum_{t=1}^{T} \sum_{k=1}^{\hat{K}} \log p_\theta\Big(\mathbf{q}_{t,k} \mid \mathbf{x}, \mathbf{q}_{<t}, \mathbf{q}_{t,<k}\Big), \quad (9)$$

where $\theta$ denotes the model parameters, $\mathbf{x}$ represents the input text token sequence, $\mathbf{q}_{<t}$ denotes all RVQ tokens from previous time steps, and $\mathbf{q}_{t,<k}$ denotes RVQ tokens from preceding layers at the same time step.

**Inference.** At inference time, we explicitly control the synthesis bitrate by selecting an inference depth $K_{\mathrm{infer}}$. The Temporal Transformer takes as input the text tokens together with the first $K_{\mathrm{infer}}$ RVQ token streams at each time step.

The Depth Transformer then autoregressively predicts only these $K_{\text{infer}}$ RVQ layers, while finer layers are omitted. Finally, the predicted RVQ tokens from the first $K_{\text{infer}}$ layers are decoded into waveforms using the TAC decoder. As TAC is trained with quantizer dropout, the decoder is inherently robust to varying effective bitrates, which aligns naturally with Progressive Sequence Dropout in the speech generation model.

**Special case.** When $p = 0$, Progressive Sequence Dropout is disabled, yielding $z = 0$ for all training samples and $\hat{K} = N_q$. In this case, the proposed method reduces exactly to the standard Temporal Transformer + Depth Transformer formulation for multi-stream autoregressive speech generation.

# 4. Experiments

## 4.1. Implementation Details

TAC consists of a large-scale causal Transformer encoder–decoder with hierarchical patching, enabling efficient streaming audio modeling. Discrete representations are learned using a 32-layer residual vector quantizer with quantizer dropout to support variable-bitrate tokenization. To encourage semantic alignment, we attach a decoder-only causal language model for audio-to-text supervision. Training is performed on approximately 3.09M hours of diverse speech, sound, and music data, using a combination of reconstruction, semantic, and adversarial objectives. All components of TAC, including the encoder, quantizer, decoder, discriminator, and decoder-only LLM, are optimized jointly in an end-to-end manner. All architectural details, optimization hyperparameters, and training schedules are provided in Appendix A.

## 4.2. Reconstruction Evaluation

We compare TAC with open-source audio tokenizers using both objective and subjective evaluation metrics across low (750–1500 bps), medium (1500–2500 bps), and high (2500–6000 bps) bitrate regimes. Table 2 summarizes the objective reconstruction results on speech, general audio, and music benchmarks.

Across all evaluated bitrate regimes, TAC achieves strong performance on speech reconstruction, outperforming prior methods at low bitrates and achieving state-of-the-art results at medium and high bitrates. On audio and music benchmarks, TAC maintains competitive performance across all evaluated bitrates, with reconstruction quality improving as bitrate increases, indicating that the model effectively benefits from increased bitrate and model capacity through joint end-to-end optimization.

Additional details on the compared open-source audio to-

kenizers, subjective evaluation results, audio/music metrics, and inference efficiency are provided in Appendices B and C.

## 4.3. Speech Generation

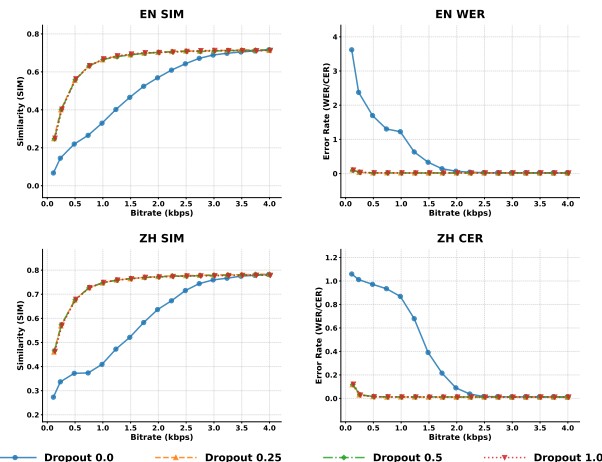

*Figure 3.* Effect of Progressive Sequence Dropout on fully autoregressive TTS across different bitrates.

**Experimental Settings.** We initialize the Temporal Transformer with the pretrained Qwen3-1.7B model (Yang et al., 2025). The Depth Transformer consists of four Transformer blocks and is randomly initialized. We train the model on a mixture of VoxBox, as introduced in SparkAudio (Wang et al., 2025), and an internal dataset, totaling approximately 200k hours of speech data. Evaluation is conducted on the Seed-TTS-Eval benchmark (Anastassiou et al., 2024). Training details are provided in Appendix D.

**Effectiveness of Progressive Sequence Dropout.** We investigate the effect of Progressive Sequence Dropout by varying the dropout probability $p \in \{0.0, 0.25, 0.5, 1.0\}$, with results summarized in Figure 3. At full bitrate, all models achieve comparable performance, exhibiting low word error rate and high speaker similarity. However, as the bitrate decreases, the model trained without dropout exhibits a much steeper degradation in similarity and word error rate. This stems from the mismatch between training and inference, as the model is trained exclusively with full RVQ depth but evaluated using truncated representations.

In contrast, models trained with Progressive Sequence Dropout are substantially more robust under reduced bitrate settings. Across different dropout probabilities ($p = 0.25, 0.5,$ and $1.0$), TTS performance remains highly consistent at each bitrate, indicating the exact dropout probability has limited impact on generation quality. Meanwhile, increasing $p$ significantly reduces GPU memory consumption during training. Therefore, we adopt $p = 1.0$ in all sub-

*Table 2.* Reconstruction quality comparison of open-source audio tokenizers on speech and audio/music data. Speech metrics are evaluated on LibriSpeech test-clean (English) and AISHELL-2 (Chinese) and reported as English/Chinese. Audio metrics are evaluated on the AudioSet evaluation subset, while music metrics are evaluated on the MUSDB dataset; values are reported as audio/music. STFT-Dist. denotes the STFT distance. Higher is better for speech metrics, whereas lower is better for audio/music metrics. $N_{\mathrm{VQ}}$ denotes the number of quantizers.

| Model | bps | Frame rate | $N_{\mathrm{VQ}}$ | Speech | | | | Audio / Music | |
| --- | --- | --- | --- | --- | --- | --- | --- | --- | --- |
| | | | | SIM↑ | STOI↑ | PESQ-NB↑ | PESQ-WB↑ | Mel-Loss↓ | STFT-Dist.↓ |
| **StableCodec** | 700 | 25 | 2 | 0.62 / 0.45 | 0.91 / 0.86 | 2.91 / 2.50 | 2.24 / 1.93 | – / – | – / – |
| **XCodec2.0** | 800 | 50 | 1 | 0.82 / 0.74 | 0.92 / 0.86 | 3.04 / 2.46 | 2.43 / 1.96 | – / – | – / – |
| **MiMo Audio Tokenizer** | 850 | 25 | 4 | 0.80 / 0.74 | 0.91 / 0.87 | 2.94 / 2.62 | 2.39 / 2.14 | **0.82** / 0.81 | 2.33 / 2.23 |
| **Higgs Audio Tokenizer** | 1000 | 25 | 4 | 0.77 / 0.68 | 0.83 / 0.82 | 3.03 / 2.61 | 2.48 / 2.14 | 0.83 / **0.80** | 2.20 / 2.05 |
| **SpeechTokenizer** | 1000 | 50 | 2 | 0.36 / 0.25 | 0.77 / 0.68 | 1.59 / 1.38 | 1.25 / 1.17 | – / – | – / – |
| **XY-Tokenizer** | 1000 | 12.5 | 8 | 0.85 / 0.79 | 0.92 / 0.87 | 3.10 / 2.63 | 2.50 / 2.12 | – / – | – / – |
| **BigCodec** | 1040 | 80 | 1 | 0.84 / 0.69 | 0.93 / 0.88 | 3.27 / 2.55 | 2.68 / 2.06 | – / – | – / – |
| **Mimi** | 1100 | 12.5 | 8 | 0.74 / 0.59 | 0.91 / 0.85 | 2.80 / 2.24 | 2.25 / 1.78 | 1.24 / 1.19 | 2.62 / 2.49 |
| **TAC (Ours)** | 750 | 12.5 | 6 | 0.82 / 0.75 | 0.93 / 0.89 | 3.14 / 2.73 | 2.60 / 2.22 | 0.86 / 0.85 | 2.21 / 2.10 |
| **TAC (Ours)** | 1000 | 12.5 | 8 | **0.88 / 0.81** | **0.94 / 0.91** | **3.38 / 2.96** | **2.87 / 2.43** | **0.82 / 0.80** | **2.16 / 2.04** |
| **DAC** | 1500 | 75 | 2 | 0.48 / 0.41 | 0.83 / 0.79 | 1.87 / 1.67 | 1.48 / 1.37 | – / – | – / – |
| **Encodec** | 1500 | 75 | 2 | 0.60 / 0.45 | 0.85 / 0.81 | 1.94 / 1.80 | 1.56 / 1.48 | 1.12 / 1.04 | 2.60 / 2.42 |
| **Higgs Audio Tokenizer** | 2000 | 25 | 8 | 0.90 / 0.83 | 0.85 / 0.85 | 3.59 / 3.22 | 3.11 / 2.73 | 0.74 / 0.70 | 2.07 / 1.92 |
| **SpeechTokenizer** | 2000 | 50 | 4 | 0.66 / 0.50 | 0.88 / 0.80 | 2.38 / 1.79 | 1.92 / 1.49 | – / – | – / – |
| **Qwen3 TTS Tokenizer** | 2200 | 12.5 | 16 | **0.95** / 0.88 | **0.96** / 0.93 | 3.66 / 3.10 | 3.19 / 2.62 | – / – | – / – |
| **MiMo Audio Tokenizer** | 2250 | 25 | 12 | 0.89 / 0.83 | 0.95 / 0.92 | 3.57 / 3.25 | 3.05 / 2.71 | **0.70 / 0.68** | 2.21 / 2.10 |
| **Mimi** | 2475 | 12.5 | 18 | 0.89 / 0.76 | 0.94 / 0.91 | 3.49 / 2.90 | 2.97 / 2.35 | 1.10 / 1.06 | 2.45 / 2.32 |
| **TAC (Ours)** | 1500 | 12.5 | 12 | 0.92 / 0.86 | 0.95 / 0.93 | 3.64 / 3.27 | 3.20 / 2.74 | 0.77 / 0.74 | 2.08 / 1.96 |
| **TAC (Ours)** | 2000 | 12.5 | 16 | **0.95 / 0.89** | **0.96 / 0.94** | **3.78 / 3.46** | **3.41 / 2.96** | 0.73 / 0.70 | **2.03 / 1.90** |
| **DAC** | 3000 | 75 | 4 | 0.74 / 0.67 | 0.90 / 0.88 | 2.76 / 2.47 | 2.31 / 2.07 | 0.86 / 0.83 | 2.23 / 2.10 |
| **MiMo Audio Tokenizer** | 3650 | 25 | 20 | 0.91 / 0.85 | 0.95 / 0.93 | 3.73 / 3.44 | 3.25 / 2.89 | 0.66 / 0.65 | 2.17 / 2.06 |
| **SpeechTokenizer** | 4000 | 50 | 8 | 0.85 / 0.69 | 0.92 / 0.85 | 3.05 / 2.20 | 2.60 / 1.87 | – / – | – / – |
| **Mimi** | 4400 | 12.5 | 32 | 0.94 / 0.83 | 0.96 / 0.94 | 3.80 / 3.31 | 3.43 / 2.78 | 1.02 / 0.98 | 2.34 / 2.21 |
| **Encodec** | 4500 | 75 | 6 | 0.86 / 0.75 | 0.92 / 0.91 | 2.91 / 2.63 | 2.46 / 2.15 | 0.91 / 0.84 | 2.33 / 2.17 |
| **DAC** | 6000 | 75 | 8 | 0.89 / 0.84 | 0.95 / 0.94 | 3.75 / 3.57 | 3.41 / 3.20 | **0.65 / 0.63** | 1.97 / 1.87 |
| **TAC (Ours)** | 3000 | 12.5 | 24 | 0.96 / 0.92 | **0.97 / 0.96** | 3.90 / 3.64 | 3.61 / 3.20 | 0.69 / 0.66 | 1.98 / 1.84 |
| **TAC (Ours)** | 4000 | 12.5 | 32 | **0.97 / 0.93** | **0.97 / 0.96** | **3.95 / 3.71** | **3.69 / 3.30** | 0.68 / 0.64 | **1.96 / 1.82** |

sequent experiments to maximize training efficiency while maintaining comparable synthesis quality.

**Comparison with Open-Source TTS Systems.** We evaluate the performance of our TAC-based fully autoregressive (AR) TTS system against a comprehensive suite of open-source models. These baselines encompass three major paradigms: (i) *cascaded systems* (e.g., AR+NAR), (ii) *purely non-autoregressive systems*, and (iii) *prior purely autoregressive systems* based on discrete or continuous representations. Detailed descriptions and categorizations of these baseline systems are provided in Appendix E.

As shown in Table 3, TAC-TTS significantly outperforms previous discrete fully autoregressive models, particularly in speaker similarity (SIM). Moreover, our method achieves competitive performance compared to recent state-of-the-art systems such as IndexTTS2, MaskGCT, and VoxCPM, with

all systems maintaining very low word error rates (WER), typically below 2%.

Notably, TAC-TTS achieves the highest speaker similarity scores on Seed-TTS-Eval for both English and Chinese among the compared open-source models. This demonstrates that scaling TAC, as a unified discrete interface, effectively captures fine-grained acoustic characteristics required for high-quality, zero-shot speech generation. Appendix D.4 further reports bilingual subjective CMOS/SMOS evaluation, where TAC-TTS obtains the best SMOS in both English and Chinese among the compared systems.

### 4.4. Speech Understanding

In addition to speech generation, we further evaluate the speech understanding capability of TAC by applying it to downstream LLM-based ASR and comparing against repre-

*Table 3.* Comparison with open-source TTS systems on Seed-TTS-Eval. Bitrate control indicates whether a TTS system allows explicit specification of the synthesis bitrate at inference time. For FlexiCodec-TTS, bitrate is controlled by switching the frame rate of the autoregressive model. For TAC-TTS, bitrate is controlled by specifying the number of RVQ tokens generated by the Depth Transformer.

| TTS Systems | Bitrate Control. | Seed-EN | | Seed-ZH | |
|---|---|---|---|---|---|
| | | WER↓ | SIM↑ | CER↓ | SIM↑ |
| Cascade (AR+NAR / NAR+NAR) | | | | | |
| MaskGCT | ✗ | 2.62 | 71.7 | 2.27 | 77.4 |
| FireRedTTS | ✗ | 3.84 | 46.0 | 1.51 | 63.5 |
| CosyVoice2 | ✗ | 3.09 | 65.9 | 1.38 | 75.7 |
| Qwen2.5-Omni | ✗ | 2.72 | 63.2 | 1.70 | 75.2 |
| CosyVoice3-1.5B | ✗ | 2.22 | **72.0** | 1.12 | **78.1** |
| IndexTTS2 | ✗ | 2.23 | 70.6 | 1.03 | 76.5 |
| FlexiCodec-TTS | ✓ | 2.63 | 65.7 | - | - |
| GLM-TTS | ✗ | **1.91** | 68.1 | **0.89** | 76.4 |
| NAR / Continuous AR | | | | | |
| F5-TTS | ✗ | 2.00 | 67.0 | 1.53 | 76.0 |
| VibeVoice | ✗ | 3.04 | 68.9 | 1.16 | 74.4 |
| VoxCPM | ✗ | **1.85** | **72.9** | **0.93** | **77.2** |
| Discrete AR | | | | | |
| Llasa | ✗ | 2.97 | 57.4 | 1.59 | 68.4 |
| SparkTTS | ✗ | 1.98 | 58.4 | 1.20 | 67.2 |
| OpenAudio-s1-mini | ✗ | 1.94 | 55.0 | 1.18 | 68.5 |
| HiggsAudio-v2 | ✗ | 2.44 | 67.7 | 1.50 | 74.0 |
| FireRedTTS2 | ✗ | 1.95 | 66.5 | **1.14** | 73.6 |
| **TAC-TTS (Ours)** | ✓ | **1.89** | **73.1** | 1.23 | **78.5** |

sentative open-source state-of-the-art speech understanding models; detailed results are provided in Appendix F. Beyond speech, we also evaluate TAC tokens in general-audio and music understanding/generation settings; these additional downstream results are provided in Appendix G.

# 5. Analysis

## 5.1. End-to-End Optimization Makes TAC a Scalable Audio Tokenizer

A key goal of TAC is *scalability*—the ability to continuously improve reconstruction quality with increased training budget. Although TAC consists of multiple adversarial components, its optimization strategy is critical for enabling such scalability. We compare **full end-to-end** optimization with the **partial** protocol used in prior works (Wu et al., 2023; Li et al., 2025c; Gong et al., 2025; Zhang et al., 2025b), where the encoder and quantizer are frozen while the decoder and discriminator are optimized. As shown in Figure 4, end-to-end training yields sustained improvements across all metrics without early saturation. In contrast, partial opti-

mization plateaus early, as freezing components restricts the model's ability to refine representations. These results demonstrate that **end-to-end optimization is crucial for scaling TAC effectively** with increased computation and capacity.

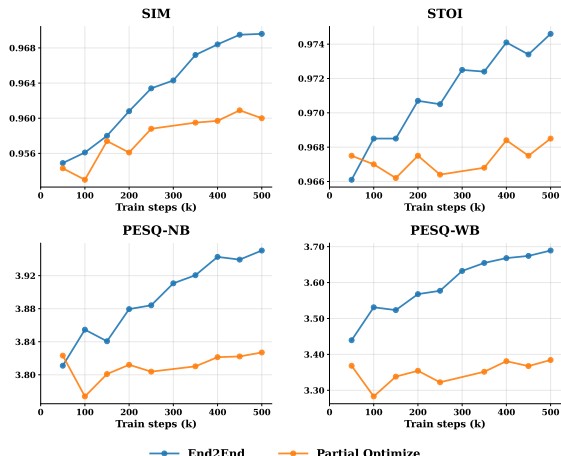

*Figure 4.* Comparison between full end-to-end optimization and partial (stage-wise) optimization for TAC.

## 5.2. Joint Scaling of Parameters and Quantization Capacity Maximizes Fidelity

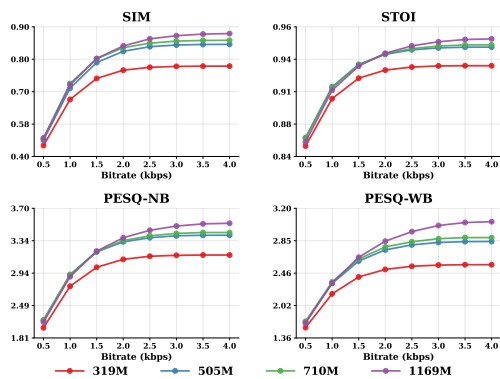

*Figure 5.* Scaling behavior of TAC reconstruction performance with respect to bitrate and model parameters.

We examine how TAC scales with model size. Following Section 3, we jointly optimize all components while varying the hidden dimension (256, 384, 512, 768)—totaling 319M, 505M, 710M, and 1169M combined encoder–decoder parameters, respectively. Throughout, the quantizer is held constant at 32 layers and 12.5 Hz.

Figure 5 shows that increasing the parameter count improves reconstruction quality across 0.5–4 kbps. While the 1169M model benefits most from high bitrates, smaller versions saturate early. Notably, at low bitrates, the 1169M model can underperform smaller models operating at higher bitrates, indicating that bitrate–rather than parameter count–becomes

the primary bottleneck. These findings reveal that **parameter scaling and quantization depth are fundamentally co-dependent**. Neither can be scaled effectively in isolation, as system performance is governed by the narrowest bottleneck. Optimal scaling thus requires a synchronized expansion of both model parameters and quantization capacity within an end-to-end framework.

## 6. Related Works

Related works on audio tokenizers, audio generation models, and audio language models are provided in Appendix I.

## 7. Conclusion

In this paper, we introduced **TAC**, a fully end-to-end Transformer-based discrete audio tokenizer that serves as a unified interface for autoregressive audio language modeling. By jointly optimizing the encoder, quantizer, and decoder within a causal architecture, TAC achieves state-of-the-art reconstruction performance among open-source audio tokenizers, and its discrete tokens support strong performance in speech generation and speech understanding. Additional general-audio and music downstream experiments further support the broader applicability of the learned tokens, while leaving a full study of all audio domains and training recipes to future work.

## Impact Statement

This paper presents work whose goal is to advance the field of machine learning, specifically in discrete audio representation and modeling. High-quality audio tokenization and zero-shot speech generation can also increase the risk of misuse, including voice spoofing, impersonation, and deepfake-style content generation. These risks motivate careful model release practices, watermarking or provenance mechanisms, and consent-aware use of speaker prompts in downstream applications.

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

# A. More Details of TAC

## A.1. Architecture

The encoder and decoder of TAC each consist of 68 causal Transformer blocks with a 10 s sliding-window attention, enabling efficient streaming inference. To progressively reduce the sequence length, the encoder inserts patchify operations (Dosovitskiy, 2020; Okamoto et al., 2023) at the input stage and after layers 12, 24, and 36, with patch sizes of 240/2/2/2, respectively. Since patchify operations modify the feature dimensionality, a linear projection is applied after each patchify stage to map the hidden states to the corresponding Transformer block dimension. This design maps raw 24 kHz waveforms to a low frame rate of 12.5 Hz.

The encoder is composed of four stages with hidden dimensions of 768, 768, 768, and 1280, respectively. These stages contain 12, 12, 12, and 32 Transformer blocks. For each stage, the feed-forward network (FFN) dimension is set to four times the corresponding hidden dimension. Multi-head self-attention uses 12, 12, 12, and 20 attention heads for the four stages, respectively. All Transformer blocks employ rotary positional embeddings (RoPE).

The decoder mirrors the encoder architecture in a fully causal manner. For a patch size $p$, inverse patchify symmetrically reshapes the hidden states from $R^{B \times (D \cdot p) \times L}$ to $R^{B \times D \times (L \cdot p)}$, restoring temporal resolution stage by stage until the 24 kHz waveform is reconstructed. Both the encoder and decoder contain approximately 0.8B parameters and are trained from scratch.

Discrete tokenization is performed using a 32-layer residual vector quantizer (RVQ). Each layer uses a codebook of size 1024 with factorized vector quantization (latent dimension 8) (Kumar et al., 2023) and L2-normalized codes. Quantizer dropout with probability 1.0 is applied during training to enable variable-bitrate tokenization.

To encourage semantically structured discrete representations, we attach a 0.5B decoder-only causal language model (Yang et al., 2024a) for audio-to-text supervision, which autoregressively predicts text conditioned on the quantizer outputs. The audio-to-text tasks include ASR, multi-speaker ASR, and audio captioning.

For adversarial training, we employ a multi-period discriminator (Défossez et al., 2022) and a complex STFT discriminator (Kumar et al., 2023). All components—encoder, quantizer, decoder, semantic head, and discriminators—are optimized jointly in an end-to-end manner.

## A.2. Dataset and Optimization

We train on approximately 3.09M hours of speech, sound, and music data, covering both clean and in-the-wild recordings, and mixing audio-only and paired (audio, text) samples.

Table 4 summarizes the training-pool composition from both the audio-domain and supervision perspectives. For sam-

*Table 4.* Composition of the TAC tokenizer training pool. The audio-domain and supervision views are complementary descriptions of the same training pool. In the table, Web denotes web-crawled general audio, MS-ASR denotes multi-speaker ASR transcripts, and ASR denotes ASR transcripts.

| Category | Hours | Note |
|---|---|---|
| **Audio domain** | | |
| Speech | 2.208M | Main training source |
| Sound | 30k | Non-speech audio |
| Web | 850k | Mixed speech/sound/music |
| **Text supervision** | | |
| MS-ASR | 8k | Paired speech transcripts |
| ASR | 2.86M | Single-speaker and general ASR |
| Caption | 10k | Audio caption supervision |
| No text | 210k | Reconstruction-only audio |

ples with available transcriptions or captions, we apply an auxiliary audio-to-text training objective, while audio-only samples are used without text supervision. Open datasets use their original transcripts or captions. Web-crawled audio is segmented into clips of at most 30 s and labeled by our internal pipeline; all audio is converted to 24 kHz mono without additional data augmentation. We optimize both the generator and discriminators using AdamW (Loshchilov & Hutter, 2017) optimizer and conduct training in bfloat16 (bf16) precision. The generator is trained with a learning rate of $1 \times 10^{-4}$ and a weight decay of 0.01, while no weight decay is applied to the discriminators. The loss weights are set to $\lambda_{\text{sem}}=20$, $\lambda_{\text{rec}}=15$, $\lambda_{\text{cmt}}=0.25$, $\lambda_{\text{code}}=1.0$, $\lambda_{\text{adv}}=1.0$, and $\lambda_{\text{feat}}=2.0$.

## A.3. Training Schedule

Due to computational constraints, we adopt a two-stage training schedule to improve training efficiency: non-adversarial pretraining without discriminator-related losses for 520k steps (batch size 1536, approximately 5 hours of audio per batch), followed by adversarial finetuning for 500k steps (batch size 768). All modules are optimized end-to-end without pretrained encoders or semantic teachers (Hsu et al., 2021; Radford et al., 2023; Zhang et al., 2023b; Défossez et al., 2024; Ye et al., 2025a).

# B. More Details on Evaluation of Audio Tokenizers

## B.1. Reconstruction Evaluation Protocol

We evaluate the reconstruction quality of TAC and open-source audio tokenizers across three domains: *speech*,

*sound*, and *music*.

**Objective evaluation.** For speech reconstruction, we conduct evaluations on LibriSpeech test-clean (English) (Panayotov et al., 2015) and AISHELL-2 (Chinese) (Du et al., 2018). We report speaker similarity (SIM), computed as the cosine similarity between speaker embeddings extracted from the original and reconstructed audio using a pretrained speaker verification model[1]. In addition, we report short-time objective intelligibility (STOI) (Taal et al., 2010) and perceptual evaluation of speech quality (PESQ) (Rix et al., 2001).

For sound and music reconstruction, following prior work (Kumar et al., 2023), we evaluate on the AudioSet evaluation subset (Gemmeke et al., 2017) and MUSDB (Rafii et al., 2017). We report mel-spectrogram distance and short-time Fourier transform (STFT) distance as objective metrics.

**Subjective evaluation.** In addition to objective metrics, we conduct a crowd-sourced listening test based on the MUSHRA protocol (Series, 2014). In this test, each listener rates the perceptual quality of reconstructed audio samples on a 1–100 scale.

For tokenizers that support variable bitrate decoding, we report results at multiple bitrates to characterize reconstruction quality across different bitrate regimes.

### B.2. Results of Subjective Evaluation

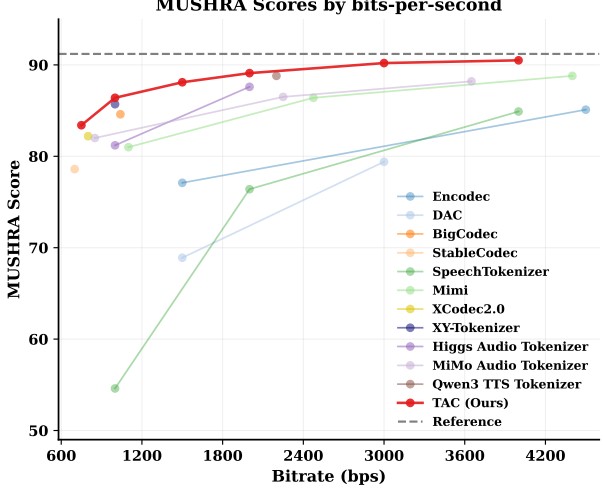

*Figure 6.* Comparison of the MUSHRA subjective test results between open-source audio tokenizers and TAC.

We conduct subjective evaluations on speech data to compare TAC with open-source audio tokenizers. For tokenizers

that support variable bitrates, we report subjective scores at multiple bitrates. The results are shown in Figure 6.

Overall, TAC achieves strong and consistent performance across a wide range of bitrates, indicating high perceptual quality in reconstructed speech. For Encodec, DAC, and SpeechTokenizer, the subjective scores are competitive at higher bitrates but degrade noticeably at lower bitrates. In contrast, audio tokenizers designed for a specific target bitrate (e.g., BigCodec, XCodec 2.0, XY-Tokenizer, and the Qwen3 TTS Tokenizer) perform well at their respective training bitrates, where their perceptual quality is competitive with TAC at comparable bitrates.

Overall, these results demonstrate that TAC provides a scalable and robust tokenizer for general audio, enabling high-fidelity compression and reconstruction of speech, sound, and music across a wide range of bitrates.

### B.3. Additional Audio and Music Metrics

Table 5 reports additional objective and subjective audio/music metrics in the representative 2500–6000 bps range. These results complement the mel-spectrogram and STFT-distance metrics in Table 2.

### B.4. Baseline Audio Tokenizers

In this section, we provide additional implementation and configuration details for the baseline audio tokenizers reported in Table 1. Unless otherwise specified, for models based on vector quantization, the target bitrate is controlled during evaluation by truncating residual vector quantization (RVQ) codes to the first several layers.

**TS3-Codec.** TS3-Codec (Wu et al., 2024) is a CNN-free Transformer codec designed for simple streaming speech reconstruction. It differs from TAC in scope and setting: TS3-Codec focuses on speech at a fixed bitrate, while TAC uses RVQ at 12.5 Hz, supports controllable bitrate, and is trained on broader speech, sound, and music data with semantic supervision. Because the TS3-Codec training code is not public, Table 6 uses the reported TS3-Codec results and should be interpreted as a limited reference rather than a fully controlled comparison.

**Encodec.** We evaluate the official causal EnCodec model operating at 24 kHz for monophonic audio[2] (Défossez et al., 2022) and it contains approximately 14 M parameters.

**DAC (Descript Audio Codec).** DAC (Kumar et al., 2023) is a neural audio codec designed for high-fidelity waveform reconstruction using carefully engineered discriminators and

---

[1] https://github.com/microsoft/UniSpeech/tree/main/downstreams/speaker_verification

[2] https://huggingface.co/facebook/encodec_24khz

*Table 5.* Additional audio and music evaluation in the representative 2500–6000 bps range. A and M denote general audio and music, respectively. SDR, SI-SDR, and MUSHRA are higher-is-better metrics.

| Model | bps | SDR-A ↑ | SDR-M ↑ | SI-SDR-A ↑ | SI-SDR-M ↑ | MUSHRA-A ↑ | MUSHRA-M ↑ |
|---|---|---|---|---|---|---|---|
| MiMo-Audio-Tokenizer | 3650 | -13.14 | -14.94 | 0.02 | 0.02 | 85.1 | 79.6 |
| Mimi | 4400 | 3.38 | 7.38 | 4.56 | 6.73 | 80.2 | 70.8 |
| Encodec | 4500 | 3.38 | 6.16 | 4.79 | 6.29 | 81.5 | 82.5 |
| DAC | 6000 | 1.57 | 4.25 | 1.39 | 3.15 | 83.5 | 86.8 |
| **TAC (Ours)** | 4000 | **4.63** | **7.50** | **5.70** | **7.44** | **87.4** | **89.8** |

*Table 6.* Comparison with TS3-Codec on LibriSpeech test-clean using reported TS3-Codec results. The comparison is not fully controlled because TS3-Codec training code is not public.

| Model | bps | SIM ↑ | STOI ↑ | PESQ-WB ↑ |
|---|---|---|---|---|
| TS3-Codec | 850 | 0.68 | 0.91 | 2.23 |
| **TAC (Ours)** | 750 | **0.82** | **0.93** | **2.60** |
| **TAC (Ours)** | 2000 | **0.95** | **0.96** | **3.41** |

improved vector quantization strategies. We use the official 24 kHz monophonic model for evaluation. The released checkpoint contains approximately 74 M parameters.

**SpeechTokenizer.** We adopt the official `speechtokenizer_hubert_avg` model[3], which is trained on monophonic speech at 16 kHz (Zhang et al., 2023b). SpeechTokenizer distills HuBERT representations using the first layer of residual vector quantization, enabling effective disentanglement of speech information, and further supports a unified speech language model (USLM). The model contains approximately 103.67 M parameters.

**Mimi.** We evaluate the official Mimi codec[4] (Défossez et al., 2024). Mimi operates on monophonic audio at 24 kHz and produces discrete audio tokens at a frame rate of 12.5 Hz, while supporting streaming encoding and decoding.

**BigCodec.** We use the authors' released checkpoint with the default 16 kHz monophonic configuration. Big-Codec (Xin et al., 2024) employs a single vector quantization (VQ) codebook with a size of 8,192 and produces discrete tokens at an 80 Hz frame rate. The model contains approximately 159 M parameters.

**Stable Codec.** For Stable Codec, we use the released `stable-codec-speech-16k-base` check-

point[5], which operates on monophonic speech at 16 kHz. Stable Codec (Parker et al., 2024) adopts a residual finite scalar quantization (RFSQ) bottleneck. Following the official recommendation, we apply the `1x46656_400bps` and `2x15625_700bps` FSQ bottleneck preset during evaluation. The base checkpoint contains approximately 953 M parameters.

**XCodec2.0.** XCodec2.0 is a semantically enhanced speech codec that incorporates a pre-trained speech encoder (Chung et al., 2021). We use the authors' released checkpoint[6] and follow the official inference pipeline. XCodec2.0 encodes 16 kHz monophonic audio into discrete tokens at a 50 Hz frame rate using a single-layer vector quantizer. The released checkpoint contains approximately 822 M parameters.

**Higgs Audio Tokenizer.** We evaluate the released `Higgs-audio-v2-tokenizer` checkpoint[7] (BosonAI, 2025), which operates on monophonic audio at 24 kHz. The checkpoint used in our experiments contains approximately 201 M parameters.

**MiMo Audio Tokenizer.** MiMo-Audio-Tokenizer (Zhang et al., 2025b) is designed to support both waveform reconstruction and downstream language modeling. The tokenizer jointly optimizes semantic and reconstruction objectives on a large-scale corpus, reportedly exceeding 11 million hours of audio. In our evaluation, we use the official released checkpoint[8]. The model contains approximately 1.2 B parameters.

**Qwen3 TTS Tokenizer.** Qwen3-TTS-Tokenizer (Hu et al., 2026) is the discrete speech tokenizer used in Qwen3-TTS for speech generation and streaming text-to-speech. We

---

[3] https://huggingface.co/fnlp/ SpeechTokenizer/tree/main/speechtokenizer_ hubert_avg
[4] https://huggingface.co/kyutai/mimi
[5] https://huggingface.co/stabilityai/ stable-codec-speech-16k-base
[6] https://huggingface.co/HKUSTAudio/ xcodec2
[7] https://huggingface.co/bosonai/ higgs-audio-v2-tokenizer
[8] https://huggingface.co/XiaomiMiMo/ MiMo-Audio-Tokenizer

evaluate the released tokenizer checkpoint[9] on monophonic audio at 24 kHz. The tokenizer encodes waveforms into discrete tokens at a frame rate of 12.5 Hz and contains approximately 170 M parameters.

**XY-Tokenizer.** XY-Tokenizer (Gong et al., 2025) is designed to mitigate the semantic–acoustic conflict at ultra-low bitrates by jointly modeling semantic and acoustic information using two encoders. We evaluate the officially released checkpoint[10]. XY-Tokenizer encodes 16 kHz monophonic audio into discrete tokens at a 12.5 Hz frame rate using an 8-layer RVQ (codebook size 1,024). Quantizer dropout is disabled in the released model. The tokenizer contains approximately 519 M parameters.

## C. Efficiency Analysis

We evaluate reconstruction-time efficiency on LibriSpeech test-clean. Table 7 compares TAC with open-source tokenizers, and Table 8 reports TAC scaling variants. Although TAC is larger than many lightweight codecs and has higher real-time per-frame latency, its batched RTF remains competitive. This reflects a practical trade-off of the homogeneous Transformer design: it favors high-concurrency packed-sequence inference, while smaller CNN-style codecs such as Mimi can be preferable for small-batch streaming latency.

*Table 7.* Streaming inference efficiency on LibriSpeech test-clean. RTF is measured at batch size 16, while realtime latency is measured as encode+decode one frame at batch size 1 for causal models. Module rows report TAC encoder/quantizer and decoder separately.

| Model | RTF ↓ | Latency (ms) ↓ |
|---|---|---|
| StableCodec | 0.017527 | – |
| XCodec2.0 | 0.021808 | – |
| MiMo-Audio-Tokenizer | 0.001332 | – |
| Higgs Audio Tokenizer | 0.002758 | – |
| SpeechTokenizer | 0.001573 | – |
| XY-Tokenizer | 0.004300 | – |
| Mimi | **0.001056** | **13.028** |
| DAC | 0.005010 | – |
| Encodec | 0.001382 | – |
| Qwen3 TTS Tokenizer | 0.003555 | 25.631 |
| **TAC (Ours)** | 0.001254 | 105.811 |
| TAC Enc.+Quant. | 0.000753 | 55.857 |
| TAC Dec. | 0.000504 | 49.954 |

[9] https://huggingface.co/Qwen/Qwen3-TTS-Tokenizer-12Hz

[10] https://huggingface.co/fdugyt/XY_Tokenizer

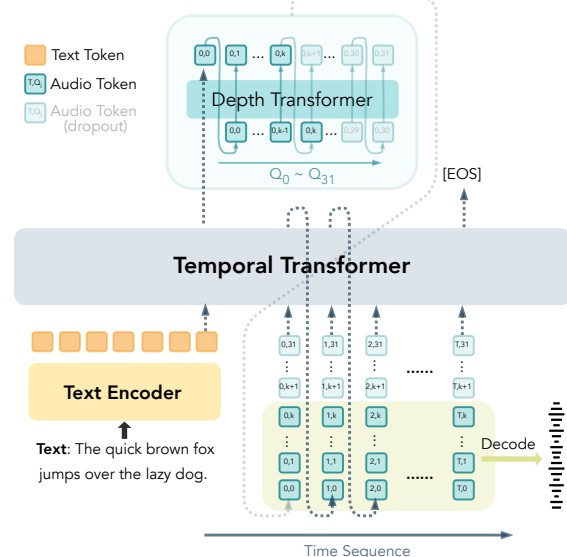

*Figure 7.* Architecture of bitrate controllable audio modeling. During training, Progressive Sequence Dropout randomly truncates the number of active RVQ layers. During inference, when decoding with a fixed depth $k$, only the first $k$ RVQ tokens are provided as input at each time step, and the Depth Transformer autoregressively predicts only these $k$ tokens, while finer RVQ layers are omitted.

## D. More Details of Bitrate Controllable Speech Generation

### D.1. Architecture

The Temporal Transformer is initialized from Qwen3-1.7B (Yang et al., 2025). The Depth Transformer is randomly initialized and consists of 4 Transformer blocks with a hidden size of 1536 and an FFN dimension of 8960.

### D.2. Training Details

We adopt a global batch size of 1.35M tokens, including both text tokens and speech tokens, where speech tokens are counted at a frame rate of 12.5 Hz. During training, text tokens and speech tokens are concatenated along the temporal dimension to form the input sequence for the TTS model. FlashAttention-2 (Dao, 2023) is used to accelerate training. All models are trained using the AdamW optimizer with a peak learning rate of $2 \times 10^{-4}$. For ablation studies, models are trained for 50k steps, while the final models are trained for 200k steps.

### D.3. Inference Details

At inference time, to maintain consistency with the training procedure, we synthesize speech in a continuation-based manner. Given a prompt audio with its transcription and a target text to be synthesized, we concatenate the prompt transcription tokens $x_{prompt}$, the target text tokens $x_{syn}$, and the prompt audio tokens $s_{prompt}$ into a single input

*Table 8.* Reconstruction-time efficiency of TAC variants on one H100 GPU with LibriSpeech test-clean. RTF is measured at batch size 128, while GFLOPs and per-frame latency are measured at batch size 1.

| Model | GFLOPs | Encode RTF ↓ | Decode RTF ↓ | Encode Latency (ms) ↓ | Decode Latency (ms) ↓ |
|---|---|---|---|---|---|
| 319M | 151.0932 | 0.000233 | 0.000113 | 54.758 | 49.328 |
| 505M | 240.6406 | 0.000245 | 0.000133 | 55.715 | 50.592 |
| 710M | 341.3023 | 0.000264 | 0.000154 | 55.254 | 49.715 |
| 1169M | 569.1365 | 0.000273 | 0.000199 | 54.861 | **48.782** |
| **TAC (1780M)** | 681.7405 | 0.000294 | 0.000220 | 55.857 | 49.954 |

sequence. The TTS model then autoregressively predicts the speech tokens corresponding to the target text. Finally, the predicted speech tokens are decoded into waveforms using the TAC decoder.

### D.4. Subjective Evaluation

Table 9 reports bilingual subjective TTS evaluation. TAC-TTS achieves the best SMOS in both English and Chinese and the best or tied-best CMOS among the compared systems, supporting the objective Seed-TTS-Eval results in Table 3.

*Table 9.* Bilingual subjective evaluation of TTS systems. CMOS and SMOS are higher-is-better metrics for comparative mean opinion score and speaker similarity mean opinion score, respectively.

| Model | EN CMOS ↑ | EN SMOS ↑ | ZH CMOS ↑ | ZH SMOS ↑ |
|---|---|---|---|---|
| MaskGCT | 0.06 | 3.82 | 0.08 | 3.83 |
| CosyVoice2 | **0.08** | 4.02 | 0.08 | 4.05 |
| IndexTTS2 | 0.07 | 3.94 | 0.06 | 4.01 |
| OpenAudio-s1-mini | 0.06 | 4.00 | 0.09 | 3.95 |
| **TAC-TTS (Ours)** | **0.08** | **4.07** | **0.10** | **4.11** |

## E. More Details on Baseline Text-to-Speech Systems

We compare our TAC-TTS system with a wide range of open-source text-to-speech (TTS) models. These models can be broadly categorized into three groups.

The first group consists of cascaded TTS systems that employ multiple generative models, such as AR+NAR or NAR+NAR architectures. Representative examples include MaskGCT (Wang et al., 2024), FireRedTTS (Xie et al., 2025), CosyVoice2 (Du et al., 2024), Qwen2.5-Omni (Xu et al., 2025a), CosyVoice3 (Du et al., 2025), IndexTTS2 (Zhou et al., 2025a), FlexiCodec-TTS (Li et al., 2025b), and GLM-TTS (Cui et al., 2025).

The second group includes purely non-autoregressive TTS systems, such as F5-TTS (Chen et al., 2025b).

The third group comprises prior fully autoregressive TTS models based on either discrete or continuous repre-

sentations, including Llasa (Ye et al., 2025b), Spark-TTS (Wang et al., 2025), OpenAudio-s1 (OpenAudio, 2024), HiggsAudio-v2 (BosonAI, 2025), FireRedTTS2 (Xie et al., 2025), DiTAR (Jia et al., 2025), and VoxCPM (Zhou et al., 2025b).

TAC-TTS adopts a purely autoregressive architecture based on discrete tokens to perform zero-shot TTS in an end-to-end manner, directly generating speech from text without relying on predefined intermediate representations, such as semantic tokens (Hsu et al., 2021; Du et al., 2024). Moreover, TAC-TTS supports variable-bitrate speech generation through Progressive Sequence Dropout.

## F. Speech Understanding on TAC

*Table 10.* ASR performance comparison.

| | Model Size | EN-WER ↓ | ZH-CER ↓ |
|---|---|---|---|
| Whisper-Large-v3 | 1.5B | 2.90 | 5.80 |
| Voxtral Small-24B | 24B | 1.53 | 13.80 |
| FireredASR-AED | 1.1B | 1.93 | 3.00 |
| Qwen2-Audio-Base | 7B | 1.74 | 3.08 |
| Baichuan-Audio-Base | 7B | 3.02 | 3.87 |
| Step-Audio-Chat | 130B | 3.11 | 3.60 |
| Qwen2.5-Omni | 7B | 2.37 | 2.56 |
| Kimi-Audio | 7B | 1.28 | 2.56 |
| **TAC-ASR (Ours)** | 1.7B | **2.96** | **3.44** |

We explore the capability of TAC for speech understanding tasks. Specifically, we investigate whether TAC tokens can be directly used as inputs to a large language model (LLM) for automatic speech recognition (ASR), in order to evaluate the alignment between TAC and text as well as the information preservation of the discrete speech representation.

We adopt Qwen3-1.7B (Yang et al., 2025) as the backbone LLM. To enable speech understanding, we initialize a set of 32 speech tokens in the vocabulary and directly feed the discretized TAC speech tokens into the LLM. For each speech frame, the tokens along the RVQ dimensions are summed and treated as a single input embedding to the LLM. The model is then trained in a fully autoregressive manner to predict the corresponding text sequence given the

speech token inputs.

The model is trained on an internal dataset consisting of approximately 2 million hours of paired (audio, text) data. We use a global batch size of 1M tokens and train the model for 200k steps with a warmup of 4k steps. The Adam optimizer is adopted with a peak learning rate of $5 \times 10^{-5}$. All experiments are conducted without any additional alignment or auxiliary supervision beyond the standard ASR objective.

We evaluate the trained TAC-based ASR model on both English and Chinese benchmarks. For English, we report word error rate (WER) on the LibriSpeech `test-clean` set (Panayotov et al., 2015). For Chinese, we report character error rate (CER) on the AIShell-2 iOS subset (Du et al., 2018). We compare our model with a range of previous open-source ASR systems and speech-language models (Radford et al., 2023; Liu et al., 2025; Xu et al., 2025c; Chu et al., 2024; Li et al., 2025c; Huang et al., 2025; Xu et al., 2025a; Ding et al., 2025), as summarized in Table 10.

As shown in Table 10, TAC-ASR achieves competitive performance across both English and Chinese benchmarks. These results suggest that TAC tokens retain sufficient linguistic content and exhibit good alignment with text, enabling effective speech understanding when directly consumed by an LLM. We believe TAC-ASR can be further improved by scaling up paired training data and model capacity.

## G. General-Audio and Music Downstream Evaluation

To evaluate downstream behavior beyond speech, we train TAC-based general-audio understanding and generation models on 3000 hours sampled from AudioCaps (Kim et al., 2019), AudioSet (Gemmeke et al., 2017), AudioStock, and FreeSound, and evaluate on the AudioCaps test subset. Table 11 shows that TAC is competitive to slightly better on captioning metrics and achieves a much lower FAD for generation than the compared tokenizers.

*Table 11.* General-audio downstream evaluation on the AudioCaps test subset. BLEU-3, CIDEr-D, and SPIDEr evaluate captioning and are higher-is-better; FAD and FD evaluate generation and are lower-is-better.

| Model | BLEU-3 ↑ | CIDEr-D ↑ | SPIDEr ↑ | FAD ↓ | FD ↓ |
|---|---|---|---|---|---|
| Mimi | 0.08 | **0.12** | 0.09 | 13.08 | **0.61** |
| MiMo-Audio-Tokenizer | 0.08 | 0.10 | 0.09 | 13.19 | 0.68 |
| **TAC (Ours)** | **0.09** | **0.12** | **0.10** | **8.77** | 0.68 |

We further evaluate music downstream understanding and generation on MusicCaps (Agostinelli et al., 2023). Because high-quality paired music-caption data are limited, we train on 3000 hours of internal music data whose captions are generated with Qwen3-Omni-30B-A3B-Instruct (Xu et al., 2025b). Table 12 reports the results. These experiments provide direct music downstream evidence, but the training data source and caption-generation pipeline should be considered when interpreting the scope of the conclusion.

## H. Additional Ablations and Scope

### H.1. Initialization and Semantic-Objective Ablations

Table 13 shows that TAC-TTS remains strong without Qwen3 initialization, suggesting that the tokenizer contribution is not solely explained by the pretrained temporal backbone. Tables 14–16 report semantic-objective ablations. The semantic loss remains active during adversarial training; the before/after comparison indicates a mild reconstruction–semantics trade-off rather than semantic collapse. Initializing the 0.5B semantic-loss LLM has only a small effect on both reconstruction and LLM-ASR metrics in this setting.

### H.2. Limitations and Scope

Our evidence supports a strong scaling trend for TAC under the presented Transformer-based training recipe, but it does not establish a universal scaling law over all tokenizer architectures, datasets, or optimization schedules. Cross-model comparisons are also constrained by differences in released checkpoints, training data, and unavailable baseline recipes. The downstream evidence is strongest for speech, with additional general-audio and music experiments included above; broader music and open-ended audio generation settings remain important future work.

## I. Related Works

### I.1. Discrete Audio Tokenizers

Discrete audio tokenizers aim to encode continuous audio waveforms into sequences of discrete tokens and reconstruct audio signals from these tokens. Most existing methods adopt an RVQGAN-style framework, which employs an encoder–quantizer–decoder architecture combined with adversarial training to achieve high-fidelity audio reconstruction (Zeghidour et al., 2021; Défossez et al., 2022; Kumar et al., 2023; Défossez et al., 2024).

SoundStream (Zeghidour et al., 2021) introduces *quantizer dropout*, enabling a single tokenizer to support variable bitrate reconstruction. TAC also uses quantizer dropout at the tokenizer level, while Progressive Sequence Dropout applies the same prefix-truncation principle to downstream autoregressive generation so that one TTS model can synthesize at multiple bitrates without retraining separate generators. Encodec (Défossez et al., 2022) further improves reconstruction quality by incorporating a multi-scale STFT (MS-STFT) discriminator to capture audio structures at different

*Table 12.* Music downstream evaluation on MusicCaps. BLEU, CIDEr-D, and SPIDEr evaluate captioning and are higher-is-better; FAD and FD evaluate generation and are lower-is-better.

| Model | BLEU-1 ↑ | BLEU-2 ↑ | BLEU-3 ↑ | CIDEr-D ↑ | SPIDEr ↑ | FAD ↓ | FD ↓ |
|---|---|---|---|---|---|---|---|
| Mimi | 0.26 | 0.13 | 0.08 | 0.07 | 0.08 | 6.94 | 0.60 |
| MiMo-Audio-Tokenizer | 0.24 | 0.11 | 0.06 | 0.05 | 0.07 | 8.82 | 0.61 |
| **TAC (Ours)** | **0.27** | **0.14** | **0.09** | **0.11** | **0.11** | **6.88** | **0.45** |

*Table 13.* Effect of Qwen3-1.7B initialization for TAC-TTS on Seed-TTS-Eval.

| Setting | EN WER ↓ | EN SIM ↑ | ZH CER ↓ | ZH SIM ↑ |
|---|---|---|---|---|
| Loaded | 1.89 | **73.1** | 1.23 | **78.5** |
| Not loaded | **1.68** | 72.8 | **1.16** | **78.5** |

*Table 15.* Effect of initializing the semantic-loss LLM on reconstruction metrics on LibriSpeech test-clean.

| Setting | SIM ↑ | STOI ↑ | PESQ-NB ↑ | PESQ-WB ↑ |
|---|---|---|---|---|
| Not load LLM | **0.94** | **0.96** | 3.80 | 3.36 |
| Load LLM | **0.94** | **0.96** | **3.82** | **3.38** |

*Table 14.* LLM-based ASR performance before and after adversarial training. The model is trained and evaluated on LibriSpeech splits.

| Setting | dev-clean ↓ | test-clean ↓ | test-other ↓ |
|---|---|---|---|
| Before adv. | **8.25** | **9.23** | **18.84** |
| After adv. | 9.15 | 10.33 | 22.53 |

*Table 16.* Effect of initializing the semantic-loss LLM on LLM-ASR WER. Models are trained on LibriSpeech train and evaluated on dev/test splits.

| Setting | dev-clean ↓ | test-clean ↓ | test-other ↓ |
|---|---|---|---|
| Not load LLM | 8.39 | 9.11 | 20.50 |
| Load LLM | **8.17** | **9.04** | **19.75** |

temporal resolutions. DAC (Kumar et al., 2023) simplifies the training process via factorized vector codes and employs complex STFT discriminators at multiple time scales to enhance phase modeling. Other acoustic codecs, including BigCodec (Xin et al., 2024), Stable-Codec (Parker et al., 2024) and TS3-Codec (Wu et al., 2024), focus on improving reconstruction quality under extremely low bitrates.

Beyond reconstruction fidelity, recent studies have explored injecting semantic information into audio tokenizers to better support downstream generative and understanding tasks. A common approach is knowledge distillation from pretrained teacher models (Hsu et al., 2021; Chung et al., 2021; Chen et al., 2022). SpeechTokenizer (Zhang et al., 2023b), Mimi (Défossez et al., 2024), and Qwen3 TTS Tokenizer (Hu et al., 2026) align the encoder and quantizer representations with self-supervised speech models through distillation objectives. In contrast, XCodec2.0 (Ye et al., 2025a), Higgs Audio Tokenizer (BosonAI, 2025), Dual Codec (Li et al., 2025a), and SAC (Chen et al., 2025a) directly initialize the tokenizer encoder using pretrained SSL or ASR models, thereby reducing the difficulty of semantic modeling.

A scale-driven approach introduces semantic information into audio tokenizers through large-scale audio–text supervision. Methods such as Baichuan Audio Tokenizer (Li et al., 2025c), XY-Tokenizer (Gong et al., 2025), and MiMo Audio Tokenizer (Zhang et al., 2025b) leverage audio-to-text

tasks and massive paired datasets, enabling the tokenizer to implicitly learn rich semantic representations while maintaining high-fidelity reconstruction.

Despite these advances, it remains unclear what characteristics make an audio tokenizer truly suitable for native audio language models. We argue that such a tokenizer should minimize handcrafted priors and architectural constraints, and instead adopt a simple and scalable design. Our goal is to obtain an audio tokenizer that is well aligned with the modeling needs of audio language models by scaling up both computation and data and training the tokenizer in an end-to-end manner.

### I.2. Audio Generation

Audio generation models have witnessed rapid progress in recent years (Kreuk et al., 2022; Borsos et al., 2023; Liu et al., 2023; Huang et al., 2023), largely driven by the combination of discrete audio representations (Chung et al., 2021; Défossez et al., 2022; Zhu et al., 2025) and large-scale language modeling (Kaplan et al., 2020; Achiam et al., 2023). A dominant paradigm is to perform generation in a compressed acoustic space, where audio is represented by sequences of discrete tokens produced by neural audio codecs (Défossez et al., 2022; 2024), and generation is formulated as a language modeling problem.

AudioLM (Borsos et al., 2023) proposes a hierarchical gen-

eration strategy that decomposes audio generation into three stages: semantic modeling, coarse acoustic modeling, and fine acoustic modeling. By combining representations from self-supervised speech models (Chung et al., 2021) with neural codec tokens, AudioLM achieves high-quality audio generation with strong long-term consistency. VALL-E (Wang et al., 2023) introduces a hybrid autoregressive (AR) (Radford et al., 2018) and non-autoregressive (NAR) (Devlin et al., 2019) architecture for speech synthesis, and demonstrates that scaling training data to tens of thousands of hours leads to the emergence of in-context learning capabilities for speech generation. Tortoise-TTS (Betker, 2023) further explores expressive text-to-speech by combining autoregressive sequence modeling with diffusion-based (Ho et al., 2020) refinement, enabling multi-voice and highly expressive synthesis.

Along this line, an important trend is the move toward end-to-end audio generation (Agostinelli et al., 2023; Liu et al., 2023; Liao et al., 2024; Ning et al., 2025; Peng et al., 2025), where a single generative model directly produces audio tokens, rather than cascading multiple generative models (e.g., BERT-style or GPT-style language models, or diffusion-based generative models) in a multi-stage pipeline. This simplification substantially reduces system complexity and error propagation across stages, while also improving training stability and inference efficiency.

In the context of discrete token-based generation, Music-Gen (Agostinelli et al., 2023) systematically studies different multi-sequence modeling patterns and finds that the delay pattern enables a single autoregressive model to perform both text- and melody-conditioned music generation. More recent systems such as Moshi (Défossez et al., 2024) adopts a combination of temporal transformers and depth transformers to efficiently model long audio sequences, and further leverage streaming audio tokenizers to significantly reduce inference latency, enabling faster and more responsive audio generation.

Beyond discrete tokenization, there is also a growing body of work on audio generation based on continuous representations. These approaches augment auto-regressive large language models with local diffusion transformers (DiT) (Liu et al., 2024; Jia et al., 2025; Peng et al., 2025; Zhou et al., 2025b), enabling the auto-regressive model to directly generate continuous latent representations and capture fine-grained acoustic details without explicit discretization.

Overall, modern audio generation research is converging toward scalable, end-to-end architectures that tightly couple representation learning and generation. This trend highlights the increasing importance of well-designed audio tokenizers that are not only faithful in reconstruction quality, but also compatible with the architectural choices and scaling properties of audio language models (Zhang et al.,

2023b; Défossez et al., 2024; Yuan et al., 2025; Zhang et al., 2025b).

**I.3. End-to-End Audio Language Models**

End-to-end audio language models (Zhang et al., 2023a; Nguyen et al., 2025; Défossez et al., 2024; Zeng et al., 2024; Li et al., 2025c; Zhao et al., 2025; Zhang et al., 2025b) aim to unify speech understanding, generation, and reasoning within a single large-scale model, moving beyond conventional three-stage pipelines that decompose speech processing into ASR, text-based language modeling, and TTS. By directly modeling audio representations using language modeling objectives, these systems aim to equip large language models with native audio understanding and generation capabilities.

Early efforts in this direction include SpeechGPT (Zhang et al., 2023a), which is among the first large-scale models to support end-to-end speech interaction. SpeechGPT leverages discrete speech representations derived from self-supervised speech encoders (Hsu et al., 2021) and scales training on large amounts of cross-modal data, enabling large language models to acquire intrinsic conversational abilities across speech and text modalities. Subsequent works such as Spirit-LM (Nguyen et al., 2025), GLM4-Voice (Zeng et al., 2024), and MOSS-Speech (Zhao et al., 2025) further improve speech–text alignment by scaling up speech–text interleaved data, demonstrating that tightly coupled multimodal pretraining is critical for robust end-to-end speech understanding and generation.

More recent systems push this paradigm to significantly larger scales. Models such as Kimi-Audio (Ding et al., 2025) and Qwen3-Omni (Xu et al., 2025b) expand training data to hundreds of thousands or even millions of hours of audio, leading to substantially improved robustness in complex and diverse audio scenarios. These results suggest that end-to-end audio language models benefit strongly from data scaling, similar to trends observed in text-only large language models (Kaplan et al., 2020; Henighan et al., 2020; Achiam et al., 2023).

An emerging line of work explores end-to-end audio language modeling based on information-preserving or near-lossless audio representations (Défossez et al., 2022; 2024). Moshi (Défossez et al., 2024) employs a multi-stream speech-to-speech Transformer together with a streaming audio tokenizer to enable full-duplex spoken dialogue, achieving low-latency, highly responsive, and human-like interactions. MiMo-Audio (Zhang et al., 2025b) further demonstrates that scaling training data to the order of 100 million hours allows end-to-end audio language models to exhibit emergent few-shot in-context learning capabilities in audio, highlighting the strong interaction between tokenizer design, data scale, and model capacity.

Overall, these studies highlight the central role of audio tokenizers in end-to-end audio language models. Similar to text tokenizers for LLMs, an audio tokenizer is expected to provide a native discrete interface that scales effectively with autoregressive modeling. Accordingly, our goal is to develop a unified, fully end-to-end trained audio tokenizer built from homogeneous causal Transformers, supporting predictable scaling with data and model capacity while minimizing handcrafted constraints.

