# OpenReview forum: "Scaling Transformers for End-to-End Discrete Audio Tokenization"
_ICML.cc/2026/Conference — ICML 2026 regular_

### Official Review · Reviewer_pRHS · 2026-03-02

**Soundness:** 4
**Presentation:** 4
**Significance:** 3
**Originality:** 1
**Overall Recommendation:** 5
**Confidence:** 4

**Summary:**

The paper proposes TAC, a Transformer-based Audio Tokenizer built from causal Transformer blocks and demonstrates the scalability of neural audio codec.

**Compliance With Llm Reviewing Policy:**

Affirmed.

**Ethical Review Concerns:**

While this paper proposes neural audio codec and text-to-speech models that may introduce significant negative societal impacts, such as deepfake misuse, these issues are not discussed in the Impact Statement section.

**Ethical Review Flag:**

Flag this paper for an ethics review.

**Ethics Expertise Needed:**

["Other Expertise"]

**Final Justification:**

Dear Authors

Thank you for your response. The concerns I raised have been fully addressed.

I noticed that the models used in the ablation study were trained with less training steps. The performance and scalability of neural audio codec are promising.

I will increase my score to 5 (Accept).

Thank you for dedicating your time to address my concern again.

Sincerely,

Reviewer pRHS

**Key Questions For Authors:**

Please provide streaming inference details (such as latency and RTF) for each model and module.

**Limitations:**

No. This paper proposes neural audio codec and text-to-speech models that may introduce significant potential negative societal impacts, such as deepfake misuse.

**Strengths And Weaknesses:**

The paper demonstrates the scalability of an RVQ-based neural audio codec with end-to-end training from scratch. While previous works require pre-trained modules such as self-supervised representation models, TAC is optimized in an end-to-end manner using reconstruction, GAN, and LM-based objectives. The large-scale neural audio codec achieves promising performance across all metrics.

However, many important details are missing.

It would be helpful to provide the model size of each model in Table 2 for a fair comparison. TAC appears to have significantly more parameters, so the reported gains are not surprising.

The dataset details are missing, which makes the comparison potentially unfair. To address this, I recommend training baseline models ( Qwen3-TTS tokenizer) on the same dataset.

**[CNN-free]**

The paper claims a CNN-free Transformer model as a key contribution. However, prior work [1] has already introduced a CNN-free Transformer neural audio codec. TAC appears to be largely an RVQ-based variant of [1]. Additionally, the manuscript does not verify the superiority of the CNN-free architecture, as there is no ablation study examining architectural design choices.

[1] Wu, Haibin, et al. "TS3-Codec: Transformer-Based Simple Streaming Single Codec." Proc. Interspeech 2025. 2025.

**[End-to-End Training and Dataset]**

The paper states that end-to-end training is essential. In this context, it is important to provide detailed information about the training dataset, including transcription and caption usage. Without these details, the results are difficult to reproduce.

**[Waveform Patchify]**

WaveNeXt [2] first introduced the patchify strategy for waveform generation. Please add the reference.
[2] Okamoto, Takuma, et al. "WaveNeXt: ConvNeXt-based fast neural vocoder without iSTFT layer." 2023 IEEE Automatic Speech Recognition and Understanding Workshop (ASRU). IEEE, 2023.

Furthermore, the patchify operation is not illustrated in Figure 2, which may confuse readers.

**[Audio/Music Evaluation]**

The paper only reports Mel-loss and STFT-distance for audio and music evaluation. These metrics alone are insufficient to demonstrate the superiority of the model.

**[Subjective Evaluation]**

It would be beneficial to include the MUSHRA results (currently in Figure 6) in the main manuscript.

**[Overall]**

While I acknowledge the product-level audio quality of the proposed model, I do not find sufficient novelty in this work. Moreover, the numerous missing details undermine the overall quality of the paper.

---

> ### Author Rebuttal · Authors · 2026-03-30
>
> Thank you for the detailed comments. We agree the original submission should have provided more implementation detail and used more careful wording.
>
> **W1.Model size**
>
> We agree that model sizes should be shown more explicitly. The parameter counts of the compared audio tokenizers are already listed in Appendix B.3, and we will also add them directly into Table 2 in the revision.
>
> At the same time, we do not attribute TAC's gains only to size: Figure 4 supports end-to-end optimization, Figure 5 supports joint scaling of tokenizer components, and the new matched-compute evidence further suggests that structure matters beyond raw parameter count.
>
> **W2.Dataset/retraining**
>
> Please refer to our response to Reviewer fRHp(`W1`) for the dataset and preprocessing details.
>
> On baseline retraining, we agree that retraining strong baselines on the same data would be ideal; however, for `Qwen3-TTS-Tokenizer`, this is currently difficult because the training code is not public and several key recipe details, such as the distillation loss, discriminator usage, and loss weights, are not specified in the technical report. We plan to reproduce it in the next stage as much as the available information allows.
>
> **W3.CNN-free,TS3-Codec**
>
> We agree we should avoid implying TAC is the first CNN-free tokenizer. Our intended claim is narrower: TAC is an end-to-end scalable Transformer tokenizer with strong empirical scaling behavior across a wide bitrate range.
>
> Relative to TS3-Codec, TAC differs in scope and scale: it supports `125-4000 bps` while TS3-Codec uses a fixed bitrate, is trained on broader speech / sound / music data while TS3-Codec focuses on speech, uses semantic guidance for downstream usability, and scales to `1.6B` parameters. At comparable bitrate, TAC is also substantially stronger in reconstruction (on `Librispeech test clean`):
>
> ||bps|SIM|STOI|PESQ-WB|
> |---|---:|---:|---:|---:|
> |TS3-Codec|850|0.68|0.91|2.23|
> |TAC|750|0.82|0.93|2.60|
> |TAC|2000|0.95|0.96|3.41|
>
> TS3-Codec utilizes a single codebook at a fixed frame rate (50hz), whereas TAC employs an RVQ architecture at a much lower frame rate (12.5Hz), which fundamentally makes TAC more suitable for seamless integration with autoregressive LLMs.
>
> For the matched-compute architectural evidence, please refer to our response to Reviewer nuw1(`W1/Q2`): pure Transformer is clearly better than CNN and comparable to CNN+Trm, while being architecturally simpler, more suitable for high-concurrency serving, and more favorable for scale-up.
>
> **W4.Reproducibility**
>
> Please refer to our response to Reviewer fRHp(`W1`); in the revision, we will make the training-pool composition, supervision types, and preprocessing pipeline explicit so the end-to-end setting is reproducible.
>
> **W5.WaveNeXt/patchify**
>
> Thank you for pointing this out. We will add the `WaveNeXt` citation and clarify patchify / inverse-patchify in Figure 2. On the decoder side, inverse patchify is symmetric to encoder-side patchify: it unpacks each patch back to the original temporal layout.
>
> **W6.Audio/music eval**
>
> We agree. For this reason, we additionally compared TAC against strong baselines on `SDR`, `SI-SDR`, and subjective quality for audio / music. In the table below, `A` denotes Audio and `M` denotes Music.
>
> We show the representative `2500-6000 bps` range below; in this range, TAC is consistently stronger:
>
> ||bps|SDR-A|SDR-M|SI-SDR-A|SI-SDR-M|MUSHRA-A|MUSHRA-M|
> |---|---:|---:|---:|---:|---:|---:|---:|
> |MiMo-Audio-Tokenizer|3650|-13.14|-14.94|0.02|0.02|85.1|79.6|
> |Mimi|4400|3.38|7.38|4.56|6.73|80.2|70.8|
> |Encodec|4500|3.38|6.16|4.79|6.29|81.5|82.5|
> |DAC|6000|1.57|4.25|1.39|3.15|83.5|86.8|
> |TAC|4000|4.63|7.50|5.70|7.44|87.4|89.8|
>
> We will move these additional audio / music metrics into the main paper.
>
> **W7.MUSHRA**
>
> Thank you for the reminder. We will move this point into the main paper.
>
> **Q1.Inference Details**
>
> We evaluated RTF(bs=16) and Latency(encode+decode 1 frame,bs=1) on LibriSpeech test-clean. We report realtime latency only for causal models.
>
> ||RTF|Lat(ms)|
> |---|---|---|
> |StableCodec|0.017527|-|
> |XCodec2.0|0.021808|-|
> |MiMo|0.001332|-|
> |Higgs|0.002758|-|
> |SpeechTokenizer|0.001573|-|
> |XY-Tokenizer|0.004300|-|
> |Mimi|0.001056|13.028|
> |DAC|0.005010|-|
> |Encodec|0.001382|-|
> |Qwen3TTS|0.003555|25.631|
> |TAC(Ours)|0.001254|105.811|
>
> TAC modules:
> Enc+Quant: RTF 0.000753, Lat 55.857
> Dec: RTF 0.000504, Lat 49.954
>
> Analysis: Smaller tokenizers excel in realtime latency(e.g.,Mimi). However, despite having ~1B parameters, TAC achieves competitive batched RTF(0.001254). This highlights a key advantage of our homogeneous Transformer design: it allows highly efficient sequence packing under high-concurrency settings, avoiding the padding overhead typical of heterogeneous CNN-based architectures.
>
>
> **L1.Misuse**
>
> We agree and will expand the impact statement to explicitly discuss misuse risks from high-quality speech generation and speaker similarity, including deepfake-style abuse and spoofing concerns.

---

> > ### Author Rebuttal · Reviewer_pRHS · 2026-04-03
> >
> > ## **UPDATE (08. Apr.)**
> >
> > Dear Authors
> >
> > Thank you for your response. The concerns I raised have been fully addressed.
> >
> > I noticed that the models used in the ablation study were trained with less training steps. The performance and scalability of neural audio codec are promising.
> >
> > I will increase my score to 5 (Accept).
> >
> > Thank you for dedicating your time to address my concern again.
> >
> > Sincerely,
> >
> > Reviewer pRHS
> >
> > ---
> >
> > Thank you for your response. However, my main concerns regarding novelty have still not been fully addressed.
> >
> > Based on the architectural comparison and the TS3-Codecs results, the performance gains appear to come primarily from scaling up the model size.
> >
> > The paper already includes an ablation study on model size (319M, 505M, 710M, and 1,169M), but the comparison is made only between TS3-Codecs (203M) and TAC (1.8B).
> >
> > As I suggested, it would be important to include TAC results at different model sizes for a fair comparison. In addition, while the largest model in the paper is reported as having 1,169M parameters, the rebuttal describes it as 1.8B. Which is correct?

---

> > > ### Author Response · Authors · 2026-04-07
> > >
> > > Thank you for the follow‑up. We respond concisely below.
> > >
> > > **Q1. Gains appear to come primarily from scaling up model size.**
> > > We agree that scaling model size contributes significantly to performance gains; however, our key finding is that scaling must be applied jointly to all components (encoder, quantizer, decoder) under end‑to‑end optimization to be effective. We agree parameter count is important, but performance is governed by *system‑level capacity*. While parameter count is an important factor, our results suggest that performance is also influenced by how capacity is distributed and jointly optimized across components. Our contribution is the **end‑to‑end joint scale‑up strategy**, which we clarify as:
> > > 1. **Quantizer capacity:** For a fixed model size, increasing RVQ capacity improves reconstruction; Figure 5 shows clear gains from higher quantizer depth.
> > > 2. **End‑to‑end joint optimization:** Jointly updating all modules (encoder/quantizer/decoder/discriminator) yields a higher ceiling than staged optimization (Figure 4).
> > > Together, these imply that scaling *all components simultaneously* under an end‑to‑end framework is the most reliable way to raise the performance ceiling, rather than scaling any single part in isolation.
> > >
> > >
> > >
> > > **Q2. TS3‑Codec comparison and model‑size confusion.**
> > > **Model size:** The Figure 5 ablations use 319M / 505M / 710M / 1169M models to study scaling trends. Due to computational constraints, these ablations were trained with *less compute* than the final model (as in Fig. 2), using global batch size = 32 and 300k steps. The final model in Table 2 is 1780M; in Appendix A.3 we state the full‑scale training recipe: non‑adversarial training for 520k steps (batch size 1536), followed by adversarial finetuning for 500k steps (batch size 768). We will clarify this distinction in the final paper.
> > > **TS3‑Codec:** Its training code is not public, so we can only compare to reported metrics, which is not fully fair. We will explicitly state this limitation. We will also emphasize TAC’s different design goal: a scalable interface for downstream LLMs (ASR/TTS), with lower frame rate (12.5Hz), controllable bitrate (0.125–4 kbps), multi‑domain in‑the‑wild training (speech/sound/music), and semantic modeling for downstream tasks. TS3‑Codec focuses more on reconstruction quality, is trained on LibriLight(speech), and does not describe LLM‑oriented downstream usage. Given these mismatches, a fully fair comparison is difficult. We will revise the paper to clearly separate scaling effects from model size and to avoid potential confusion in model configurations.
> > >
> > > Reference numbers (for clarity):
> > >
> > > |Model|Params|
> > > |---|---:|
> > > |TS3‑Codec|203M|
> > > |TAC ablations (Figure 5)|319M / 505M / 710M / 1169M|
> > > |TAC (final, Table 2)|1780M|
> > >
> > > **Q3. Provide TAC results at different sizes for fair comparison.**
> > > We already report 319M/505M/710M/1169M in Figure 5 and will add training‑setting details to the appendix to avoid confusion with the final 1780M model. Representative reconstruction numbers:
> > >
> > > |Model|bps|SIM|STOI|PESQ‑WB|
> > > |---|---:|---:|---:|---:|
> > > |ablation‑319M|875|0.64|0.89|2.08|
> > > |ablation‑505M|875|0.67|0.90|2.22|
> > > |ablation‑710M|875|0.69|0.90|2.24|
> > > |ablation‑1169M|875|0.68|0.90|2.21|
> > > |ablation‑319M|1000|0.67|0.90|2.18|
> > > |ablation‑505M|1000|0.71|0.91|2.32|
> > > |ablation‑710M|1000|0.72|0.91|2.34|
> > > |ablation‑1169M|1000|0.72|0.91|2.33|
> > > |ablation‑319M|2000|0.77|0.92|2.50|
> > > |ablation‑505M|2000|0.82|0.94|2.74|
> > > |ablation‑710M|2000|0.83|0.94|2.77|
> > > |ablation‑1169M|2000|0.84|0.94|2.84|

---

### Official Review · Reviewer_fRhp · 2026-03-10

**Soundness:** 4
**Presentation:** 4
**Significance:** 3
**Originality:** 2
**Overall Recommendation:** 5
**Confidence:** 5

**Summary:**

In this work, the authors propose a purely Transformer-only, causal, and end-to-end optmization architecture for the audio codec. The authors argue that with this architecture choice, the model can be well-scaled to larger model sizes and the data. Furthermore, they apply a Progressive Sequence Dropout to control the bitrate during training, and as a result, they can generate speech with various bitrates with a single model. They also propose a pure autoregressive Transformer architecture for the TTS model on top of their tokenizer. In the ablation study, they show the effect of the end-to-end training schema, different bitrates, and the model size.

**Compliance With Llm Reviewing Policy:**

Affirmed.

**Final Justification:**

I look forward to open-sourcing the model, and I will describe the data component in the appendix. I will maintain my score.

**Key Questions For Authors:**

The paper is clearly written. I would appreciate the authors' response to points W1–W3.

**Limitations:**

yes

**Strengths And Weaknesses:**

**Strengths**

The paper is clearly written and easy to understand. Their architecture choices are technically sound; with this autoregressive and variable bitrate choice, the model can support streaming and can also control the inference speed.

The ablation study is very well evaluated. They clearly show that the end-to-end training is important, which many other previous works haven’t done. I believe this can give a clear insight to the readers. Also, showing the relationship between the model size and bitrate is interesting to me.

**Weaknesses**

W1. The main limitation would be the transparency (reproducibility). They argue that scaling data is the primary motivation and factor for achieving better performance (Sec 1, P2). However, they do not disclose the dataset information, nor do they conduct an ablation study on the data scale. I strongly believe that at least the dataset construction structure (e.g., the ratios and amounts of speech, audio, and music data, or the data augmentation methods) should be described.

W2. The novelty also seems limited. One of their "Progressive Sequence Dropout” methods is quite well known and was also introduced in the Soundstream paper. They mention this in the related work section, but could not find a clear difference. Clarifying this point would be appreciated.

W3. The novelty of the model architecture is also limited, since it is a standard architecture used in recent neural audio codecs. The semantic information injection with Decoder-only LLM seems different from related work. Still, it would also limit the data, since it requires an audio-textual description-paired dataset, which is the opposite choice to their argument that data scaling is crucial. In contrast, other similar approaches, such as Mimi Codec, use WavLM, which doesn’t require any text description.

---

> ### Author Rebuttal · Authors · 2026-03-30
>
> Thank you for the positive assessment and the concrete suggestions. We will use the rebuttal and revision to make the paper clearer on reproducibility and claim scope.
>
> **W1. On reproducibility and dataset transparency**
>
> We agree the dataset description should have been much clearer. Our tokenizer training data total about `3.09M` hours, including `2.208M` speech hours, `30k` sound hours, and `850k` web-crawled general-audio hours; the supervision types include about `8k` multi-speaker ASR-transcript hours, `2.86M` ASR-transcript hours, `10k` caption hours, and `210k` hours without text.
>
> We separate the two views below:
>
> |Data type|Hours|Note|
> |---|---:|---|
> |speech|2.208M|main training source|
> |sound|30k|non-speech audio|
> |web-crawled general-audio|850k|mixed speech/sound/music; exact internal ratio is not currently available|
>
> |Text type|Hours|
> |---|---:|
> |multi-speaker ASR transcript|8k|
> |ASR transcript|2.86M|
> |caption|10k|
> |no text|210k|
>
> Open datasets use their original transcripts / captions; the crawled data are segmented into clips of at most `30s` and labeled by our internal pipeline; all audio is converted to `24kHz` mono, with no additional augmentation.
>
> To ensure reproducibility and benefit the community, we will open-source the pre-trained TAC checkpoints, the downstream TAC-TTS models, and the complete training/inference pipeline upon acceptance.
>
> **W2. On the distinction from prior quantizer dropout**
>
> We agree dropout itself is not new in audio codecs, and we will revise the wording to be more precise. To clarify the core distinction: prior methods like SoundStream use dropout purely as a codec-level technique for compression. In contrast, our Progressive Sequence Dropout is introduced as a generation-level training strategy for autoregressive LLMs.
>
> Our intended claim is narrower: TAC not only uses progressive truncation during tokenizer training, but also carries the same variable-bitrate mechanism into downstream autoregressive LLM/TTS, so one downstream model can synthesize at different bitrates by predicting only the first `k` RVQ layers, with a direct tradeoff among quality, latency, and generation cost. Because both TAC and downstream TTS support variable bitrate, unlike prior approaches that require training multiple TTS models to support different bitrates, we only need one TTS model, which greatly reduces cost.
>
> **W3. On architecture novelty and audio-text supervision**
>
> We agree our contribution is not a brand-new codec backbone in isolation. The more precise framing is that TAC scales a simple homogeneous Transformer tokenizer and shows that this route can deliver both strong reconstruction and strong downstream usability, supported by Table 2 and Figure 1 for reconstruction and Tables 3 and 4 for downstream tasks.
>
> As you noted, some prior works model semantics by distilling SSL models, e.g., SpeechTokenizer from HuBERT and Mimi from WavLM. However, such SSL models often require complex multi-stage iterative pretraining, which increases system complexity, and their robustness in harder settings such as in-the-wild or noisy-background data is often weaker than models scaled on large audio-text pairs, such as Whisper or Qwen3-Omni. Inspired by this trend, we chose to model semantics by scaling large audio-text pair data, and we believe the resulting representation is more robust.
>
> We chose audio-text supervision because large-scale `(audio,text)` data were available in our setting and this route scales naturally to downstream understanding and generation; at the same time, we agree that the current results do not fully disentangle tokenizer quality from supervision design.

---

> > ### Author Rebuttal · Reviewer_fRhp · 2026-04-03
> >
> > Thank you for your response. I look forward to open-sourcing the model, and I will describe the data component in the appendix. I will maintain my score.

---

> > > ### Author Response · Authors · 2026-04-07
> > >
> > > Thank you for your positive acknowledgement. We will clarify the data composition and preprocessing details in the appendix, and upon acceptance we will open‑source the model checkpoints and training/inference pipeline to maximize reproducibility.

---

### Official Review · Reviewer_nuw1 · 2026-03-12

**Soundness:** 2
**Presentation:** 2
**Significance:** 3
**Originality:** 2
**Overall Recommendation:** 4
**Confidence:** 4

**Summary:**

This paper introduces TAC, an end-to-end discrete audio tokenizer. Its main claim is that, rather than relying on pretrained encoders or complex hybrid designs, a single causal Transformer architecture that jointly trains the encoder, quantizer, and decoder from scratch scales more effectively. The paper demonstrates that these tokens support competitive results in pure autoregressive TTS and speech understanding, highlighting their potential as a unified interface for audio foundation models.

**Compliance With Llm Reviewing Policy:**

Affirmed.

**Final Justification:**

I thank the authors for the additional rebuttal and experiments. The new results substantially address my main concerns, particularly on the CNN comparison, the non-speech evidence, and the semantic-loss initialization effect. I still think the architectural claim should be stated more cautiously and framed as setting-specific rather than universal. With that caveat, I am now positively inclined and raise my recommendation to 4.

**Key Questions For Authors:**

1. I am not sure whether this can be interpreted as a general model scaling law. Do the authors view this as evidence of a general scaling law, or only as a TAC-specific scaling trend?
2. There is a lack of direct evidence for why end-to-end is important and why the architecture needs to be unified. I am curious whether the authors provide a comparison with distillation-based approaches, and whether the authors have directly compared against CNN-based alternatives under the same data / parameter / training setup. From the current results alone, it is difficult to disentangle whether this is a structural advantage or simply the effect of using a larger model and more data. For example, DAC is also trained end-to-end and is CNN-based, so why could it not outperform TAC if scaled to the same size and trained in the same way? What direct evidence supports the claim that CNN-based designs are intrinsically less scalable, rather than simply different design points?
3. Did the authors test stronger semantic heads or richer understanding supervision to determine whether the current semantic performance is limited by tokenizer quality or by the semantic supervision setup?
4. Since the paper positions TAC as a general audio tokenizer, are there semantic downstream results beyond speech, such as audio captioning or other general-audio understanding tasks?
5. The paper says that the encoder is patchified, but what happens when the decoder upsamples? In other words, I am curious about the implementation of inverse patchify / unpatchify.
6. During adversarial training, the semantic loss from the decoder-only 0.5B is still present, right? Also, how does the performance on semantic understanding tasks change before and after adversarial training? I am wondering whether the model loses semantic performance while focusing more on reconstruction during adversarial training.
7. In Eq. (9), how is (q_{<t}) implemented in the depth transformer? In existing depth transformers, it seems to be causal only depth-wise, while modeling each time step independently in the time direction.
8. In “Semantic Modeling via Audio-to-Text Tasks,” is the 0.5B semantic head trained from scratch? In speech generation, the temporal transformer is initialized from Qwen3-1.7B, but is the additional decoder-only model in ASR from scratch? Why did the authors use different initializations?
9. In Section 5.1, do the encoder and quantizer have the same initial weights in the full and partial settings? Also, it seems important whether these weights come from a sufficiently converged model, so how did the authors set up the experiment to ensure a fair comparison?

**Limitations:**

The paper does not meaningfully discuss limitations, and the Impact Statement is extremely brief. The authors should acknowledge the limited evidence for their strongest architectural claims, the gap between “general audio” positioning and largely speech-centric understanding evaluation, fairness issues in cross-model comparisons, and potential misuse risks related to high-quality speech generation and speaker similarity.

**Strengths And Weaknesses:**

*Soundness*

The paper convincingly shows that TAC works well and exhibits a meaningful scaling trend. However, the stronger claim that a simple, homogeneous, end to end architecture is necessary for a higher performance ceiling is not fully established. The current evidence shows that full end to end optimization helps within TAC and that scaling TAC improves reconstruction, but it does not directly show that TAC is fundamentally more scalable than CNN based, heterogeneous, or distillation based alternatives. In particular, there is no controlled comparison under matched data, parameter count, and training recipe. As a result, it is hard to separate true architectural benefits from the effects of larger models and more data. The TAC TTS results are also difficult to interpret cleanly because the tokenizer contribution is not disentangled from the strong pretrained Qwen3 1.7B initialization.

*Presentation*

The overall idea and experimental scope are strong, but several important points remain unclear. For example, the term “end to end” refers here to joint optimization of tokenizer components, yet it can easily be read as end to end training of the full downstream system. The encoder side is described fairly clearly, but the decoder side is less clear, especially the implementation of inverse patchify or unpatchify for sequence length recovery. The related work coverage is broad, but some of the most important distinctions from closely related methods are left to the appendix rather than made explicit in the main text.

*Significance*

A scalable audio tokenizer that combines strong reconstruction with downstream usability is an important building block for speech and audio foundation models. The paper is practically meaningful in its treatment of the tokenizer as an interface for LLM style or autoregressive generation, and its low frame rate tokenization, causal design, and bitrate controllable TTS are all useful contributions. The reconstruction and TTS results are especially strong. That said, the impact is better understood as establishing a reconstruction heavy but semantically usable tokenizer, rather than a universal interface for all audio understanding tasks, since semantic evidence outside speech remains limited.

*Originality*

The contribution is less about a single new mechanism and more about combining several ideas into a coherent system. A fully Transformer based tokenizer, large scale joint optimization, unified semantic and reconstruction objectives, and variable bitrate TTS through RVQ depth are not individually unprecedented, but their integration is meaningful and creative. The broader framing of audio tokenization as a simple and scalable architecture problem is also interesting. However, the novelty claims should be made more precisely.

---

> ### Author Rebuttal · Authors · 2026-03-30
>
> Thank you for the detailed feedback. We will revise the overly broad original framing.
>
> **W1.Scaling claim,architecture claim,and Qwen3 initialization**
>
> We agree the current evidence supports a strong TAC-specific scaling trend rather than a general scaling law.
>
> Matched-compute evidence under the same recipe (200k steps,global batch size=32) on `Librispeech test clean` is summarized below.
>
> ||Params(M)|GFLOPs|SIM|STOI|PESQ-WB|
> |---|---:|---:|---:|---:|---:|
> |Trm|320.91|151.89|0.69|0.91|2.37|
> |CNN+Trm|375.63|161.77|0.70|0.91|2.36|
> |CNN|67.37|162.51|0.65|0.89|2.13|
>
> This does not prove CNN designs are intrinsically unscalable in all settings, but it does show that Transformer-based modeling is important in our setting, while a homogeneous Transformer stack is simpler to scale and to serve with packed variable-length sequences.
>
> The ablation below shows our TTS remains strong even without Qwen3 initialization:
>
> ||EN WER|EN SIM|ZH CER|ZH SIM|
> |---|---:|---:|---:|---:|
> |Loaded|1.89|73.1|1.23|78.5|
> |Not loaded|1.68|72.8|1.16|78.5|
>
> **W2."End-to-end" wording and inverse patchify**
>
> We agree "end-to-end" and inverse patchify should be stated more clearly. In our paper,"end-to-end" means jointly optimizing tokenizer components,not jointly training the full downstream stack.
>
> Unpatchify simply expands one frame into multiple frames along the dimension axis.
>
> **W3.Evidence beyond speech**
>
> Please refer to our response to Reviewer axt1 (`W2`).
>
> **W4.Precision of the novelty claim**
>
> We agree the novelty claim should be more precise. Our contribution is not a single unprecedented block,but a coherent system that combines a homogeneous causal-Transformer tokenizer,large-scale joint optimization,semantic+reconstruction objectives,and bitrate-controllable autoregressive TTS into a route that is both reconstruction-strong and downstream-usable.
>
> **Q1.General scaling law or TAC-specific trend?**
>
> Our current evidence supports a strong TAC-specific scaling trend,not a general law. Establishing a general law would require a dedicated matched-setup study across multiple architectures and recipes.
>
> **Q2.Direct evidence for the "unified/end-to-end/Transformer" claim**
>
> Direct evidence comes from two places.
>
> First,Figure 4 shows that full end-to-end optimization (all modules updated jointly) has higher modeling efficiency and a higher ceiling than partial optimization.
>
> Second,the matched-compute comparison in `W1` shows that both Transformer-based variants outperform a pure CNN tokenizer under the same recipe.
>
> We did not train a distilled TAC variant. Our current setting mainly uses `(audio,text)` pairs,so we scaled model size and data size along that axis and obtained strong reconstruction plus strong downstream understanding/generation performance.
>
> **Q3.Stronger semantic heads or richer semantic supervision?**
>
> We did not test this in the current round. We think how to better add semantic guidance to audio tokenizers is a feature-design question worth exploring.
>
> **Q4.Semantic downstream results beyond speech?**
>
> Please refer to our response to Reviewer axt1 (`W2`).
>
> **Q5.How is inverse patchify/unpatchify implemented?**
>
> Please refer to `W2`.
>
> **Q6.Semantic performance before and after adversarial training**
>
> The semantic loss remains present during adversarial training. We also believe that near-lossless reconstruction is very important for audio tokenizers,which is close in spirit to the design philosophy of MiMo-Audio; accordingly,we train TAC to make reconstruction as strong as possible,and under that condition,as downstream understanding/generation data scale up,TAC still performs strongly on the tasks shown in Tables 3 and 4.
>
> LLM based ASR WER(train and test on `LibriSpeech`) before/after adversarial training is below.
>
> ||dev-clean|test-clean|test-other|
> |---|---:|---:|---:|
> |Before adv.|8.25|9.23|18.84|
> |After adv.|9.15|10.33|22.53|
>
> This indicates a mild reconstruction/semantics tradeoff rather than semantic collapse.
>
> **Q7.Depth Transformer**
>
> Please refer to our response to Reviewer axt1 (`Q3`).
>
> **Q8.Initialization**
>
> The `0.5B` semantic head used during tokenizer training is initialized from `Qwen2.5-0.5B`,while the downstream decoder-only LLM used for ASR/TTS is initialized from `Qwen3-1.7B`. They serve different roles: the smaller semantic head provides scalable audio-to-text supervision during tokenizer training,while the larger downstream model provides stronger sequence modeling for ASR/TTS.
>
> **Q9.Figure 4 fairness**
>
> Yes. The full and partial settings in Figure 4 start from the same converged non-adversarial checkpoint,so the encoder and quantizer have identical initial weights. More importantly,before branching,the training loss and evaluation metrics of that checkpoint had already converged,which makes the comparison much cleaner.
>
> **L1.Limitations/risks**
>
> We agree and will revise the paper to clarify TAC-specific scope,data/comparison limits,and misuse risks such as spoofing or deepfake-style abuse.

---

> > ### Author Rebuttal · Reviewer_nuw1 · 2026-04-03
> >
> > I thank the authors for their rebuttal. However, I still do not think the following points are fully resolved.
> > 1. In W1, the CNN baseline has similar GFLOPs but far fewer parameters, so the comparison is still confounded by both architecture and model capacity. Because of this, I do not think the current evidence is sufficient to support the stronger claim that the unified Transformer architecture is inherently better or more scalable. Figure 4 shows that full joint optimization is better than partial optimization within TAC, but it still does not clearly establish the superiority of the end-to-end unified Transformer architecture itself.
> > 2. If the evidence beyond speech, especially for music, is still limited or left for future work, then the paper’s framing should be revised accordingly. Claims such as outperforming prior codecs across speech, sound, and music seem stronger than what is currently supported. If this part is revised, then the scope and contribution of the paper may need to be reassessed accordingly.
> > 3. A less central but still relevant point is the semantic-loss LLM initialization. Since initialization effects were discussed for other components, it seems worth checking this part as well, as it could also have a nontrivial impact.

---

> > > ### Author Response · Authors · 2026-04-07
> > >
> > > Thank you for the continued feedback. We respond point‑by‑point.
> > >
> > > **Q1. CNN baseline has similar GFLOPs but far fewer parameters; evidence still confounded.**
> > > We added a *matched‑compute & matched‑parameter* CNN vs Transformer comparison. These results suggest that Transformer and CNN exhibit different trade‑offs: Transformer achieves better performance on similarity and intelligibility metrics, while CNN shows advantages in perceptual quality (PESQ) and low‑latency inference; both have similar RTF, but Transformer’s batch inference peak **GPU memory** is far lower (4.3GB vs 25.7GB), which is advantageous for high‑concurrency serving. CNN shows lower real‑time latency, suggesting it is preferable for small‑batch streaming.
> > > We will **not** claim Transformer is universally superior. Instead, we will state that *in our setting* a homogeneous Transformer is easier to scale and deploy efficiently, and yields a better overall trade‑off for reconstruction + downstream usability. In addition, in our current architecture, pushing a pure CNN to an order‑of‑magnitude higher compute via larger width/depth tends to be harder to optimize and less stable in our experiments, while Transformer is easier to optimize and scale in our experiments (e.g., TAC at 681.7405 GFLOPs achieves clear gains). From a scaling perspective, most pure CNN models remain below ~1B parameters, while Transformer‑based LLMs scale to tens or hundreds of billions (even trillions). This is one practical consideration that motivates our choice of Transformer in this work.
> > >
> > > We also note the large peak‑memory gap (≈6×) between Transformer and CNN in our matched setup. This memory difference may make it more challenging to scale CNNs to larger parameter regimes in practice: increasing CNN width/depth or batch size quickly hits hardware limits, while Transformer with packed‑sequence efficiency scales more gracefully. We therefore avoid claiming architectural superiority, and instead position our contribution as demonstrating that a homogeneous Transformer design is a viable and scalable alternative in this setting. We will revise the paper to avoid architectural over‑claims and clearly present these results as setting‑specific evidence.
> > >
> > > We emphasize that our results do not establish inherent architectural superiority, but rather highlight a favorable design point under our setting.
> > >
> > > **Evaluation setting (matched‑compute table):**
> > > - bf16 + flash_attn + packed sequences; 1x H100; LibriSpeech test‑clean.
> > > - RTF & batch‑memory measured at batch size = 16; realtime latency at batch size = 1 (1 frame per step).
> > > - Both models are causal, 24kHz, RVQ‑32, trained with the same global batch size and steps.
> > >
> > > Matched‑compute table:
> > >
> > > |Model|Params (M)|GFLOPs|SIM|STOI|PESQ‑WB|Encode RTF|Decode RTF|Batch peak mem|Enc Lat (ms)|Dec Lat (ms)|
> > > |---|---:|---:|---:|---:|---:|---:|---:|---:|---:|---:|
> > > |Trm|70.51|159.76|0.78|0.91|2.05|0.0000649|0.0000412|4.3GB|40.671|26.674|
> > > |CNN|67.37|162.51|0.65|0.89|2.13|0.0000518|0.0000498|25.7GB|23.786|6.946|
> > >
> > > **Q2. Non‑speech evidence (especially music) still limited; framing should be revised.**
> > > We added music downstream understanding and generation. Because high‑quality (music segment, caption) data are scarce, we used 3000 hours of internal music data and generated captions with Qwen3‑Omni‑30B‑A3B‑Instruct; evaluation is on MusicCaps. TAC outperforms Mimi and MiMo‑Audio‑Tokenizer on both understanding and generation metrics.
> > > We will further tighten wording to avoid “outperforms across speech/sound/music” as a blanket claim, and clearly state evidence boundaries and data limitations.
> > >
> > > Music downstream (MusicCaps):
> > >
> > > |Model|BLEU-1|BLEU-2|BLEU-3|CIDEr-D|SPIDEr|FAD|FD|
> > > |---|---:|---:|---:|---:|---:|---:|---:|
> > > |Mimi|0.26|0.13|0.08|0.07|0.08|6.94|0.60|
> > > |MiMo‑Audio‑Tokenizer|0.24|0.11|0.06|0.05|0.07|8.82|0.61|
> > > |TAC|0.27|0.14|0.09|0.11|0.11|6.88|0.45|
> > >
> > > **Q3. Semantic‑loss LLM initialization.**
> > > We ran an ablation on the semantic‑loss LLM (Qwen2.5‑0.5B initialization vs random) under the same training setting (global batch size 256, 200k steps, same data). The impact on reconstruction and semantic performance is small.
> > >
> > > Reconstruction metrics (LibriSpeech test‑clean):
> > >
> > > |Setting|SIM|STOI|PESQ‑NB|PESQ‑WB|
> > > |---|---:|---:|---:|---:|
> > > |Not load LLM|0.94|0.96|3.80|3.36|
> > > |Load LLM|0.94|0.96|3.82|3.38|
> > >
> > > Semantic metrics (LLM‑ASR WER; train on LibriSpeech train, eval on dev/test):
> > >
> > > |Setting|dev‑clean|test‑clean|test‑other|
> > > |---|---:|---:|---:|
> > > |Not load LLM|8.39|9.11|20.5|
> > > |Load LLM|8.17|9.04|19.75|
> > >
> > > We will include these tables and details in the appendix and clarify that this component is not the primary source of gains.

---

### Official Review · Reviewer_axt1 · 2026-03-13

**Soundness:** 3
**Presentation:** 2
**Significance:** 3
**Originality:** 2
**Overall Recommendation:** 4
**Confidence:** 3

**Summary:**

This paper proposes a Transformer-based universal audio tokenizer. By utilizing a unified causal Transformer architecture and joint optimization on massive datasets, the model outperforms existing mainstream open-source codecs in reconstruction performance across various audio domains (speech, ambient sound, and music) at different bitrates. To achieve consistency between training and inference for variable bitrates, the authors introduce the Progressive Sequence Dropout strategy. Additionally, based on this discrete tokenizer, the authors trained an autoregressive TTS model, demonstrating highly competitive performance. The experiments also thoroughly validate the effectiveness of scaling, proving that audio reconstruction quality improves continuously as the number of parameters increases, showing excellent architectural scalability.

**Compliance With Llm Reviewing Policy:**

Affirmed.

**Final Justification:**

The paper has strengths in terms of empirical performance, and the rebuttal added several results and analyses, including general-audio experiments, subjective TTS evaluation, and computational cost measurements, which improve the completeness of the work and address part of my concerns. I also appreciate that the authors are willing to narrow several originally overstated claims, including limiting the scaling conclusion to the Transformer setting, softening the wording around “the first purely autoregressive TTS,” and making the “universal audio tokenizer” framing more careful. These changes make the contribution more precise and better aligned with the evidence presented. That said, I still have reservations about the paper’s novelty. Overall, I think the rebuttal has partially addressed my concerns and demonstrates a constructive willingness to revise the paper’s framing and strengthen its empirical support. While I still do not view the novelty as especially strong, I am willing to raise my score to weak accept.

**Key Questions For Authors:**

1. Since TAC is explicitly designed as a universal tokenizer for speech, ambient sound, and music, why is the downstream evaluation limited to speech (TTS and ASR)? Can the authors provide additional experiments on ambient sound or music understanding/generation to fully demonstrate its effectiveness?
2. When evaluating TTS generation quality, the current tables lack subjective listening tests.
3. How are different RVQ layers within the same time step specifically arranged during autoregressive TTS training? The description of the Depth Transformer input/output logic is unclear. Does it use sequence flattening, a delay pattern, or does the Temporal Transformer predict the first layer while the Depth Transformer handles the rest?
4. For models of different sizes (319M to 1169M), can the authors provide data on computational overhead (FLOPs) and inference latency during the reconstruction stage to evaluate the practical cost of this pure Transformer architecture?

**Limitations:**

Evaluate the performance of this tokenizer on non-speech tasks to demonstrate its versatility.

**Strengths And Weaknesses:**

Strengths:

1. This paper proposes a CNN-free audio tokenizer architecture based entirely on Transformers. By discarding the inductive biases of traditional hybrid architectures, it provides a unified, scalable discrete interface for audio language models.
2. This paper provides sufficient experimental data to verify the positive impact of parameter scaling (from 319M to 1169M) on performance, strongly supporting the core claim of the model being "scalable."
3. Under identical bitrate conditions, TAC consistently outperforms existing baselines across multiple evaluation metrics.

Weaknesses:

1. Aside from replacing the underlying architecture with a pure Transformer, the core innovations (such as using Dropout to achieve variable bitrates) have been widely used in earlier audio tokenizers (SoundStream and DAC). The performance gains seem to stem more from the scaling of large-scale data and parameters rather than the introduction of entirely new mechanisms.
2. The paper claims that the tokenizer is a universal model covering speech, ambient sound, and music. However, downstream generation and understanding tasks are only validated via speech-related experiments (TTS and ASR). The lack of empirical evidence in ambient sound or music tasks weakens the rigor of the claim that it serves as a "foundation for universal audio language models."
3. The paper states, "we develop the first purely autoregressive TTS model that surpasses prior non-autoregressive and cascaded systems." This statement appears overly absolute.

---

> ### Author Rebuttal · Authors · 2026-03-30
>
> Thank you for the careful review. We agree several original claims were too broad and will revise them accordingly.
>
> **W1. On novelty, Progressive Sequence Dropout, and the role of the pure-Transformer architecture**
>
> For the PSD / dropout point, please refer to our response to Reviewer fRhp (`W2`).
>
> Regarding architecture and scaling, our main take-away is not that only one module should be enlarged, but that scaling all tokenizer components together is important (Figure 5). As we scale a homogeneous Transformer tokenizer, TAC stays strong at both low (`~1 kbps`) and higher (`~4 kbps`) bitrates, whereas prior tokenizers often focus more on one regime. For the matched-compute architectural evidence, please refer to our response to Reviewer nuw1 (`W1/Q2`).
>
> **W2. On the mismatch between the "universal audio" claim and speech-only downstream evidence**
>
> We agree that evaluating non-speech downstream tasks is a valuable addition to comprehensively demonstrate the model's capabilities. We now added TAC-based general-audio understanding and generation models, trained on `3000` hours sampled from `AudioCaps`, `AudioSet`, `AudioStock`, and `FreeSound`, and evaluated on the `AudioCaps` test subset:
>
> |Model|BLEU-3|CIDEr-D|SPIDEr|FAD|FD|
> |---|---:|---:|---:|---:|---:|
> |TAC|0.09|0.12|0.10|8.77|0.68|
> |Mimi|0.08|0.12|0.09|13.08|0.61|
> |MiMo-Audio-Tokenizer|0.08|0.10|0.09|13.19|0.68|
>
> On captioning, TAC is competitive to slightly better; on generation, TAC achieves much lower FAD with comparable FD. We will therefore narrow the wording: we now have direct downstream evidence on speech and general-audio tasks, while music downstream validation remains future work.
>
> **W3. On the overly absolute wording "the first purely autoregressive TTS model..."**
>
> We agree this phrase was too absolute. We will revise it to a narrower claim: a controllable-bitrate fully autoregressive discrete TTS system with strong objective performance relative to prior non-autoregressive and cascaded systems.
>
> **Q1. If TAC is positioned as a universal tokenizer for speech / ambient sound / music, why is downstream evaluation limited to speech?**
>
> Please refer to `W2` above. We now provide direct downstream evidence beyond speech on general-audio tasks, while music downstream validation remains future work.
>
> **Q2. Subjective TTS Eval**
>
> We agree a dedicated subjective TTS study is important, and we therefore added bilingual CMOS/SMOS results:
>
> |Model|EN CMOS|EN SMOS|ZH CMOS|ZH SMOS|
> |---|---:|---:|---:|---:|
> |MaskGCT|0.06|3.82|0.08|3.83|
> |CosyVoice2|0.08|4.02|0.08|4.05|
> |IndexTTS2|0.07|3.94|0.06|4.01|
> |OpenAudio-s1-mini|0.06|4.00|0.09|3.95|
> |TAC-TTS|0.08|4.07|0.10|4.11|
>
> TAC-TTS achieves the best SMOS in both English and Chinese (`4.07 / 4.11`), and CMOS is also best or tied-best (`0.08 / 0.10`). Thus, TAC-TTS is strong not only on objective metrics but also on subjective naturalness and speaker similarity.
>
> **Q3. How are different RVQ layers arranged within the same time step during autoregressive TTS training?**
>
> We will clarify this more explicitly in the paper. We do not use sequence flattening or an extra delay pattern. The Temporal Transformer models temporal context, while the Depth Transformer autoregressively models RVQ-depth dependencies within each frame. Concretely, if the RVQ depth is $ N_q $, the prediction of $ q_{t,k} $ depends on the text / past-audio context provided by the Temporal Transformer and on $ q_{t,<k} $ within the current frame.
>
> **Q4. What are the FLOPs and inference latency during reconstruction for models of different sizes?**
>
> We measured reconstruction-time efficiency on `1x H100` with `LibriSpeech test-clean`. `RTF` is measured at `batch size = 128`, while `FLOPs` and `latency` are measured at `batch size = 1`. Here, `latency` means the time to encode one frame or decode one frame in a streaming setting:
>
> |Model|GFLOPs|Encode RTF|Decode RTF|Encode Latency (ms)|Decode Latency (ms)|
> |---|---:|---:|---:|---:|---:|
> |319M|151.0932|0.000233|0.000113|54.758|49.328|
> |505M|240.6406|0.000245|0.000133|55.715|50.592|
> |710M|341.3023|0.000264|0.000154|55.254|49.715|
> |1169M|569.1365|0.000273|0.000199|54.861|48.782|
> |TAC(1780M)|681.7405|0.000294|0.000220|55.857|49.954|
>
> As model size increases, FLOPs rise substantially, but batched RTF changes little. We attribute this to the homogeneous pure-Transformer design: because the model contains only Transformer blocks, samples within a batch can be packed much more effectively, which is especially favorable for high-concurrency deployment and avoids the padding waste that is more common with CNN-style heterogeneous stacks. This same packed-sequence advantage is widely used in modern LLM training and serving frameworks such as Megatron-LM and vLLM.
>
> **L1. Please evaluate non-speech tasks to demonstrate tokenizer versatility**
>
> Agreed; please refer to `W2`. We have now added general-audio downstream results and will narrow the claim to speech + general audio, with music left as future work.

---

> > ### Author Rebuttal · Reviewer_axt1 · 2026-04-03
> >
> > Thank you for the detailed rebuttal and the additional experiments. I appreciate the authors’ efforts to clarify the paper and to narrow several originally overbroad claims.
> >
> > I am still not fully convinced by the explanation of novelty. In particular, the contribution around dropout now seems to be framed mainly as a training strategy for autoregressive LLMs. For a paper centered on the tokenizer itself, this innovation appears somewhat limited.
> >
> > On the positive side, I appreciate that the authors have substantially toned down several original claims: the scaling claim is now restricted to the Transformer setting, the dropout claim is narrowed to the LLM/TTS setting, the wording around “the first purely autoregressive TTS” has been softened, and the “universal audio tokenizer” framing has also been made more careful. I think these revisions make the contribution more rigorous, and I encourage the authors to reflect them clearly in the final paper.
> >
> > I also appreciate the newly added audio-related downstream results, subjective evaluation, and computational cost analysis. These additions address part of my concerns. I hope these results will be incorporated into the main paper or appendix. For the subjective evaluation, it would also be better to report confidence intervals and significance testing.
> >
> > Overall, my concerns are partially resolved. Although I still find the novelty somewhat limited, I appreciate the authors’ willingness to narrow the claims and provide additional supporting experiments. Based on this, I am willing to adjust my score somewhat more positively. Good luck.

---

> > > ### Author Response · Authors · 2026-04-07
> > >
> > > Thank you for the detailed follow‑up and for recognizing our efforts to narrow claims and add experiments. We respond concisely below.
> > >
> > > **Q1. Novelty still seems limited (dropout framed mainly for AR LLMs).**
> > > We agree that our contribution is not a new primitive module, but rather a system‑level integration and scaling study. Our goal is to demonstrate that a simple, homogeneous Transformer tokenizer, when jointly scaled across all components, can achieve strong reconstruction and downstream usability in a unified framework. PSD is important because it *naturally aligns with quantizer dropout* and enables a **single** downstream AR model to support multiple bitrates. Prior work typically requires separate downstream models for different bitrates; our setup avoids that and makes bitrate control practical without retraining new generators. We will make this positioning explicit in the final version to avoid overstating novelty.
> > >
> > > **Q2. Please reflect claim softening clearly in the final paper.**
> > > We will do so explicitly. In the final version we will:
> > > - restrict scaling conclusions to the Transformer setting,
> > > - remove absolute phrasing such as “first purely autoregressive TTS,”
> > > - narrow “universal audio tokenizer” to reflect evidence boundaries (speech + general audio + music downstream).
> > >
> > > **Q3. Incorporate new results; add statistics for subjective evaluation.**
> > > We will move the added downstream and compute‑cost results into the main paper or appendix, and we will report confidence intervals and significance testing for subjective evaluations in the final version. We added bilingual subjective TTS results and will include the full table with statistical details in the final manuscript.
> > >
> > > **Q4. Overall comments.**
> > > We appreciate your constructive stance and will ensure the revised manuscript cleanly reflects the tightened claims and added evidence.

---

### Decision · Program_Chairs · 2026-04-30

**Decision:**

Accept (regular)

**Comment:**

**Paper summary**

The paper studies how to build discrete audio tokenization in a scalable simple way to support arbitrary bitrate within the same model used for speechLLMs/audioLLMs. Authors build it using patchifying on raw wave and later to reduce frame rate, and pure causal transformer layers. To achieve consistency between training and inference for variable bitrates, the authors introduce the Progressive Sequence Dropout strategy. Authors also apply their tokenization to TTS and ASR to validate that tokenization is viable for speechLLMs/audioLLMs as the next step.

**Decision justification / recommendation**

Reviewers initially had a lot of concerns for the paper, but during rebuttal authors resolved most of them so that reviewers raised their scores having two weak accept and two accept. All reviewers agreed that the paper has comprehensive evaluation after rebuttal, and proposed tokenizer provides strong performance across bit rates, and importantly is at low frame rate (12.5Hz, which is practical). The only concern remaining across reviewers is novelty, as all proposed components and training techniques were known / used before, but reviewers agreed that despite limited novelty, demonstrating that trained from scratch pure transformer models can achieve strong performance and scale is interesting. So overall all insights are valuable and can open new directions in how we build audio tokenizers. Another missing piece for me is actually showing that tokenizer is working for speechLLM/audioLLM, as still TTS and ASR may not have enough evidence that tokenizer is well suited for language modeling. Given support from reviewers and a new view on how we should build tokenizers, I still **recommend acceptance of the paper**.

**Critical: request for the changes in final revision**

As pointed out by reviewers and promised by authors:
- there should be changes on tone down a bunch of claims in the paper as they are overstated;
- extra empirical results provided during rebuttal and discussion must be included into the paper;
- expand impact statement to discuss text-to-speech models and related to speech tokenizers potential issues.